# Influence of intraseasonal eastern boundary circulation variability on hydrography and biogeochemistry off Peru

Jan Lüdke[1], Marcus Dengler[1], Stefan Sommer[1], David Clemens[1], Sören Thomsen[2], Gerd Krahmann[1], Andrew W. Dale[1], Eric P. Achterberg[1,3], Martin Visbeck[1,3]

[1]GEOMAR Helmholtz Centre for Ocean Research Kiel, Düsternbrooker Weg 20, D-24105 Kiel, Germany

[2]LOCEAN-IPSL, IRD/CNRS/Sorbonnes Universites (UPMC)/MNHN, UMR 7159, Paris, France

[3]Kiel University, Kiel, Germany

*Correspondence to*: Jan Lüdke (jluedke@geomar.de)

Abstract. The Peruvian upwelling system is characterized by high primary productivity fuelled by the supply of nutrients in a highly dynamic boundary circulation. The intraseasonal evolution of the physical and biogeochemical properties is analysed based on shipboard observations conducted between April and June 2017 off central Peru and remote sensing data. The poleward transport in the subsurface Peru Chile Undercurrent was highly variable and strongly intensified between mid and end of May. This intensification was likely caused by a first baroclinic mode coastal trapped wave excited at the equator at about 95° W that propagated poleward along the South American coast. An intensified poleward flow increases water mass advection from the equatorial current system to the study site. The impact of the elevated advection was mostly noticed in the nitrogen cycle. Shorter transit times between the equator and the coast off central Peru led to a strong increase in nitrate concentrations, less fixed nitrogen loss to $N_2$, and a decrease in the nitrogen deficit. The study highlights the importance of alongshore advection due to coastal trapped waves for the nutrient budget and the cumulative strength of N-cycling in the Peruvian OMZ.

## 1 Introduction

The Peruvian Upwelling System (PUS) is one of the biologically most productive regions in the world's ocean resulting in economically important fish catches (e.g. Carr, 2002; Chavez et al., 2008). Located in the Eastern Tropical South Pacific (ETSP), the high surface productivity of the PUS is most pronounced within 100 km of the Peruvian coast between 4° and 16° S (Pennington et al., 2006). Equatorward winds favour upwelling throughout the year (Bakun and Nelson, 1991; Strub et al., 1998) and enable a supply of nutrients from subsurface waters to the euphotic zone, stimulating high primary productivity (Pennington et al., 2006). Beneath the upwelling system, the low oxygen waters supplied to the PUS and enhanced local oxygen consumption due to remineralization of exported organic matter lead to the development of a pronounced Oxygen Minimum Zone (OMZ). The core of the OMZ with upper and lower bounds at about 30-200 m and 400-500 m depth, respectively, is considered to be fully anoxic (e.g. Ulloa et al., 2012, Thomsen et al. 2016).

The eastern boundary circulation in the PUS is dominated by the poleward Peru-Chile Undercurrent (PCUC) occupying the upper continental slope and shelf at depths from 50 to 200 m (Fig. 1, e.g. Gunter, 1936; Chaigneau et al., 2013). It carries low-oxygen and high-nutrient Equatorial Subsurface Waters (ESSW), which constitutes the main source of the coastal upwelled waters (Brink et al., 1983; Penven et al., 2005; Silva et al., 2009; Albert et al., 2010; Chaigneau et al., 2013; Grados et al., 2018). Mean poleward velocities associated with the PCUC are between 0.05 and 0.15 ms[-1] (Chaigneau et al., 2013). The origin of its source waters is still under debate. Chaigneau et al. (2013) traced its source waters to the Equatorial Undercurrent supplying the PCUC via the Ecuador-Peru Coastal Current which flows poleward along the coast between the equator and

5° S. In contrast, a regional model analysis by Montes et al. (2010) suggests that the source waters of the PCUC originate predominately from the eastward Southern Subsurface Countercurrents south of the equator (Fig. 1). Above the PCUC, upwelling dynamics imply the existence of an equatorward geostrophic surface jet, known as the Peru Coastal Current (e.g. Hill et al., 1998; Kämpf and Chapman, 2016). Additional features of the eastern boundary circulation are the Peru-Chile Current flowing equatorward offshore of the PCUC and the Chile-Peru Deep Coastal Current flowing equatorward below the PCUC (Fig. 1, Strub et al., 1998; Penven et al., 2005; Czeschel et al., 2011; Chaigneau et al., 2013).

The PUS exhibits variability on a wide range of timescales including intraseasonal (e.g. Brink et al., 1982; Gutiérrez et al., 2008; Belmadani et al., 2012; Pietri et al., 2014, Illig et al., 2018a, 2018b), seasonal (e.g. Pizarro et al., 2002; Chaigneau et al., 2013), and interannual to decadal (Pizarro et al., 2002; Ramos et al., 2008; Graco et al., 2017). In particular, intraseasonal variability of oxygen and nutrient concentration along the Peruvian continental slope and on the shelf are known to strongly impact benthic and pelagic ecosystems within the OMZ and the upwelling region above (e.g. Gutiérrez et al., 2008; Echevin et al., 2014; Graco et al., 2017). A prominent benthic example is the biota responses to the frequency of oxygenation episodes changing from macro-invertebrates dominated biomass to bacterial mats (*Thioploca*) or when the frequency reduces (e.g. Gutiérrez et al., 2008). Furthermore, the availability of nitrate nutrients in the anoxic waters of the OMZ allows for degradation of organic matter and is thereby consumed, while the lack of nitrate nutrients favours benthic sulfide emissions and sulfide plumes in the water column (e.g. Dale et al., 2016; Schunck et al., 2013).

A key process thought to be responsible for the intraseasonal variability of the solute concentrations is variability of the eastern boundary circulation. Intraseasonal variability of the eastern boundary circulation is either forced locally by changes of the wind system above the PUS or forced remotely by variability of the wind system at the equator. Strengthening (weakening) of the local alongshore winds causes intensified (reduced) Ekman divergence close to the coast that accelerates (decelerates) the coastal surface jet (e.g. Philander and Yonn, 1982; Yoon and Philander, 1982; McCreary et al., 1987; Fennel et al., 2012). At the same time, Coastal Trapped Waves (CTWs) are excited propagating poleward to set up an alongshore pressure gradient balancing the accelerating (decelerating) alongshore flow (Yoon and Philander, 1982). Due to differences in vertical structure of the surface jet and the excited CTWs, the poleward flowing PCUC accelerates (decelerates). Likewise, variability of local wind stress curl forces variability of the poleward undercurrent through enhancing or reducing Sverdrup transport in the eastern boundary current regime (e.g. McCreary and Chao, 1985; Marchesiello et al., 2003; Junker et al., 2015; Klenz et al., 2018). Local wind-forced variability of eastern boundary poleward undercurrents have been reported from all eastern boundary upwelling regions on time scales from intraseasonal to seasonal (e.g. Allen and Smith, 1981; Marchesiello et al., 2003, Junker et al., 2015, Klenz et al., 2018).

Alternatively, intraseasonal variability of the eastern boundary circulation in the PUS may be of equatorial origin and manifested as a poleward- moving CTW as well. Variability of zonal winds in the equatorial Pacific forces equatorial Kelvin waves that propagate eastward. Upon reaching the continental margin, part of their incoming energy is transmitted poleward along the southwestern coast of South America as CTWs (e.g. Moore and Philander, 1977; Enfield et al., 1987; Rydbeck et al., 2019). Early measurement programs off Peru investigating intraseasonal alongshore current variability showed coherent CTW signals with periods of 5-50 days propagating poleward between 5° S and 15° S (Brink et al., 1980, 1983; Romea and Smith, 1983). Tide gauge data and the lack of correlated local winds suggested that the CTWs were predominately of equatorial origin. Similarly, intraseasonal CTW variability (50-day period) in the PCUC off Chile having velocity amplitudes of up to 0.7 m s$^{-1}$ were shown to have originated from wind variability in the central equatorial Pacific (Shaffer et al., 1997). Modelling efforts confirm that equatorially forced intraseasonal CTWs propagate along the eastern South American coast to as far as 30° S (Hormazabal et al., 2002; Illig et al., 2018b).

While modulating the alongshore circulation and vertical velocities, poleward propagating CTWs produce vertical displacements of the pycnocline of the order of tens of meters and sea level changes of a few centimeters (Leth and Middelton, 2006; Colas et al., 2008; Belmadani et al., 2012). The combined effect alters the horizontal and vertical supply of nutrients in

the PUS. Modelling efforts and satellite observations suggest that subsurface nutrient and chlorophyll intraseasonal variability in the PUS are mainly forced by remotely forced CTWs (Echevin et al., 2014). The simulations indicate that the shoaling and deepening of the nutricline, as well as the horizontal currents associated with the CTW induce a nutrient flux in and out of the euphotic layer, which impacts primary production. On the other hand, intraseasonal sea surface temperature (SST) variability is suggested to be mainly driven by local winds and heat fluxes while CTWs play only a minor role (Dewitte et al., 2011; Illig et al., 2014). On interannual timescales CTWs can dominate the variability in nutrient concentrations (Graco et al., 2017) and bottom water oxygen levels that can lead to strong changes in benthic ecosystems (Gutierrez et al., 2008).

Here, we use an extensive data set from a multi-cruise observational program to analyse the impact of a strongly elevated individual propagating CTW on the circulation, hydrography and nutrient concentrations in the PUS. By visiting sampling stations several times over a period of more than two months, we measured the variability of the circulation and hydrographic and biogeochemical conditions almost continuously and can distinguish the intraseasonal variability from that on shorter time scales due to other processes.

The conditions off Peru in early 2017 were further affected by anomalously warm temperatures in the upper ocean during March associated with a coastal El Niño event (e.g. Garreaud, 2018; Echevin et al., 2018). Our observations on which this analysis is based cover the declining phase of the coastal El Niño event in April and May when SST anomalies were decreasing.

## 2 Data

### 2.1 Ship data

Within the framework of the Collaborative Research Center 754 "Climate-biogeochemistry interactions in the tropical ocean", we carried out a combined physical and biogeochemical sampling program in the ETSP from April to June 2017 on R/V Meteor (Tab. 1). A regional focus during the four individual cruises of the sampling program was a transect starting at shallow waters off Callao (Peru) at about 12° S running offshore perpendicular to the coastline to water depths larger than 5000 m and >100 km offshore (Fig. 1). During the first R/V Meteor cruise M135, the 12° S section was studied at the end of the cruise on April 7 - 8. The two subsequent cruises M136 and M137 focused on benthic and pelagic work off Peru between 11° and 14° S (Sommer et al., 2019, Dengler and Sommer, 2019). Benthic lander measurements required the vessel to remain close to the section between April 18 and May 29, 2017, when repeated hydrographic and velocity measurements along the section were collected. The section was again resampled during the final cruise M138 on June 24. In this study, we analyse shipboard velocity data, hydrographic profiles as well as oxygen and nutrient concentration measurements from the repeat measurements at 12° S.

### 2.1.1 Shipboard velocity observations

During the cruises upper ocean velocities were recorded continuously using two vessel-mounted Ocean Surveyor Acoustic Doppler Current Profiler systems (vmADCP) installed on R/V Meteor. One vmADCP was operating at a frequency of 75 kHz (OS75). System configuration during the cruises varied only in depth bin settings. Depth bins of 4, 8 or 16 m were chosen depending on strength of the backscatter signal and the focus of the investigation. The second vmADCP operated at 38 kHz (OS38) which recorded depth bins of 32m while sampling an increase depth range (30-1000m) compared to the OS75 system. During post-processing, vmADCP velocities were corrected using a mean amplitude and misalignment angle determined from water-track calibration (e.g. Fischer et al., 2003). Misalignment angles derived from individual ship accelerations and turns followed a Gaussian distribution having a standard deviation of less than 0.65° for the OS75 and less than 0.75° for the OS38 (Sommer et al., 2019). A temporal trend was not detectable. The uncertainty of the misalignment angle calibration can be determined from its standard deviation divided by the square root of the number of independent estimates (e.g. Fischer et al., 2003). For our data, more than 100 independent estimates were available for each cruise, resulting in an angle uncertainty of

less than 0.1° and an associated velocity bias of less than 1 cm s$^{-1}$ for underway data. Fischer et al. (2003) suggested an accuracy of 3 cm s$^{-1}$ for hourly underway data recorded during calm conditions in the tropics.

### 2.1.2 Hydrographic observations

At the 12° S section a total of 151 hydrographic profiles were collected during the cruises M136 and M137 with a lowered SeaBird SBE 9-plus conductivity-temperature-depth (CTD) system using two pumped oxygen, temperature and conductivity
sensors each. The CTD was attached to a General Oceanics rosette with 24 Niskin bottles of 10 l to collect water samples. For the calibration of the conductivity sensor water samples were analysed with a Guildline Autosal Salinometer model 8400 B. Correction coefficients for the CTD's conductivity sensors were derived using a multiparameter fit of the Autosal conductivities against the uncalibrated CTD sensor measurements. Coefficients included an offset and factors for temperature, pressure and conductivity.

CTD oxygen sensors were calibrated against oxygen concentrations determined from bottle water samples using Winkler titration (Winkler, 1888; Grasshoff et al., 1983). Processing and calibration followed the GO-SHIP recommendations (Hood et al., 2010). Previous studies using Switchable Trace amount OXygen (STOX) sensors have shown that the core of the Peruvian OMZ is anoxic (Revsbech et al., 2009; Thomsen et al., 2016). Unfortunately, the Winkler titration method fails to accurately determine very low oxygen concentrations from bottle water samples. To assure anoxic conditions in the OMZ
core, a mean oxygen concentration offset of 2.26 µmol l$^{-1}$ was subtracted. Calibration of the salinity and oxygen sensors was performed separately for each cruise, except for M136 where the mean of the calibration of the preceding and succeeding cruises M135 and M137 was used (M136 lacked the required deep-water samples). The final post-cruise calibration of the data resulted in an accuracy for temperature, salinity, and oxygen of 0.002° C, 0.002 g kg$^{-1}$ and 1.5 µmol kg$^{-1}$, respectively.

### 2.1.3 Nutrient measurements

Water samples collected during the upcast of the CTD rosette were used to determine nutrient concentrations. Concentrations of nitrate, nitrite and phosphate were measured using a QuAAtro autoanalyzer (Seal Analytical) with the precision of 0.1 µmol l$^{-1}$, 0.1 µmol l$^{-1}$, and 0.2 µmol l$^{-1}$, respectively (Sommer et al., 2019). Ammonium concentrations were measured using a fluorimetric method (Holmes et al., 1999).

In addition, concentrations of nitrate were measured using a Satlantic Deep Submersible Ultraviolet Nitrate Analyzer (SUNA)
mounted on the CTD rosette. SUNA measurements are based on the absorbance spectra of ultraviolet light (Sakamoto et al., 2009). Data post-processing followed Karstensen et al. (2017) and Thomsen et al. (2019). In a final step, the SUNA nitrate concentrations were calibrated against the nitrate concentrations from the CTD water samples using a linear fit.

### 2.2 Additional data

To supplement our analysis of ship-board observations we used additional data sets. Sea Level Anomaly (SLA) data based on
satellite altimeter measurements were used – provided by the E.U. Copernicus Marine Environmental Monitoring Service (product: SEALEVEL_GLO_PHY_L4_REP_OBSERVATIONS_008_047). This is a level 4 dataset derived by merging all available satellite altimetry data into one gridded product. Reprocessed data from January 1, 1993 to January 1, 2018 with the release days January 15, 2018 (data before May 15, 2017) and May 16, 2018 was accessed. Additionally, sea surface temperature from the NOAA Extended Reconstructed Sea Surface Temperature, Version 5 (ERSSTv5) dataset (Huang et al.,
2017a) was analyzed. This dataset provides monthly values of SST on a 2°x2° grid based on in-situ temperature observations from several sources (Huang et al., 2017b). Finally, we analysed wind stress data from the ASCAT product of satellite scatterometer winds (Bentamy and Fillon, 2012). This product provides daily global winds and wind stress data with a resolution of 0.25°.

## 3 Methods

### 3.1 Analysis of velocity observations

The continuous velocity recording was split into segments when the ship was moving in on- or offshore directions only. The velocities were rotated to derive the alongshore component and then a mean velocity section was calculated for each segment of the cruise in 2 km bins according to offshore distance. Periods where the ship was moving slower than 1 kn were excluded. To derive the sections of alongshore velocity over longer time periods, the data from several of these segments were averaged. The presented sections were smoothed using a 2D Gaussian weighting with an influence radius of 4 km (8 m) and a cut-off of 6 km (18 m) horizontally (vertically).

### 3.2 Analysis of hydrographic and biogeochemical data

The analysis of hydrographic data is based on the TEOS10 definitions (IOC et al., 2010) and conservative temperature, absolute salinity and (potential) density were calculated with the Matlab Gibbs Seawater Toolbox (McDougall and Barker, 2011; Version 3.05). Nutrient concentrations were transformed to $\mu mol\ kg^{-1}$ to remove the effect of compressibility on vertical gradients.

#### 3.2.1 Nitrogen deficit

In suboxic environments, microbially mediated biogeochemical processes such as denitrification and anaerobic ammonium oxidation (anammox) transform biologically available nitrogen nutrients (nitrate, nitrate and ammonium) into a form unusable by most organisms. These processes lead to a loss of nitrogen nutrients (N-loss) relative to other nutrients such as phosphorus. A metric for N-loss processes is the nitrogen deficit. A deficit (or excess) in nitrogen exists when the ratio of nitrogen to phosphorus deviates from the ratio associated with the synthesis and remineralisation of organic matter, i.e. the Redfield ratio (Gruber and Sarmiento, 1997). Here, we used the formulation of Chang et al. (2010) to investigate the nitrogen deficit. It is based on the deviation from the ratio between nitrogen species and phosphate outside of the eastern tropical south Pacific OMZ. The nitrogen deficit is calculated as:

$$N_{def} = 15.8\ (PO_4^{3-} - 0.3) - (NO_3^- + NO_2^- + NH_4^+) \tag{1}$$

Where $PO_4^{3-}$, $NO_3^-$, $NO_2^-$, and $NH_4^+$ are the concentrations of phosphate, nitrate, nitrite and ammonium, respectively. With this definition, positive values of $N_{def}$ quantify the N-loss that has occurred within a certain water mass while it has remained within the Peruvian OMZ.

#### 3.2.2 Section averaging

Hydrographic and biogeochemical properties along the 12° S section analysed in this study were calculated by first interpolating the data onto common potential density surfaces. Performing subsequent analysis in density space removes the effect of internal waves which cause elevated variability of vertical displacement of properties in the water column. The profiles were then averaged in bins of 2 km according to distance from the coastline. The data were smoothed using a 2D Gaussian weighting with a density influence range (standard deviation of the Gaussian distribution) of 0.03 kg m$^{-3}$ and a cut-off range of 0.05 kg m$^{-3}$ and respectively an influence and cut-off range of 3 and 6 km within 40 km of the coast, 7 and 15 km for the distances between 40 km and 80 km offshore and 15 and 20 km for more than 80 km offshore. The decreasing scale of horizontal interpolation towards the coast was used to benefit from the increased number of profiles to preserve smaller scale features. Finally, the averaged and interpolated properties were transformed back into depth space.

### 3.3 Sea level anomaly and wind data

The SLA along the 12° S section was calculated by averaging all daily gridded SLA data between 12° and 12.5° S over the time periods used for the velocity sections and interpolating them onto the section according to the distance to the coast. Grid points closer than 30 km to the coast have been excluded. The mean SLA along the section was subtracted to focus on the cross-shore gradient. Intraseasonal variability along the equator and coastline was analysed by subtracting the mean SLA over the 25-year time series before bandpass filtering SLA using a 4th order Butterworth filter for a time window between 20 and 90 days. To follow intraseasonal SLA variability due to propagating waves along the equator and subsequently along the western South American coast, we used bandpass filtered SLA data averaged between 0.25° S and 0.25° N from the equator and two-grid-points averages from near the coast.

For investigating wind forcing of Kelvin waves at the equator, SLA and zonal wind anomaly were averaged to five-day means over 10° of longitude. Additionally, wind anomaly was averaged between 5° N and 5° S while SLA was averaged between 2° N and 2° S.

### 3.4 Theoretical coastal trapped wave structure

To interpret the observed flow variability along the Peruvian coast in terms of CTWs, the cross-shore-depth structure of CTWs was determined by considering the linear, hydrostatic, inviscid, and Boussinesq approximated equations of motion on an f-plane using local bathymetry and stratification (Brink, 1982; 1989; Illig et al., 2018a). For alongshore scales larger than cross-shore scales and horizontally uniform stratification, cross-shore-vertical mode structures (eigenfunctions) and corresponding phase velocities (eigenvalues) solutions can be obtained from the simplified set of equations by using a resonance iteration approach (Brink, 1982; Brink and Chapman, 1987). Here, we obtained the eigenfunctions and eigenvalues for the first three modes by applying a modified version of the Brink and Chapman (1987) Coastal-Trapped Wave programs as published by Brink (2018). These solutions have been used successfully in previous analyses of CTW structures in observational and model data (e.g. Brink et al., 1982; Pietri et al., 2014; Illig et al., 2018a).

Local bathymetry and stratification are required as input to Brink's (2018) set of Matlab mfiles. The cross-shore distribution of bathymetry was taken from the multi-beam echo sounder data collected during the cruises along 12° S. Echo sounder data was averaged in 5 km bins which resulted in a monotonic increase of water depth in offshore direction. Water depth lower than 5000 m were ignored. From two offshore CTD profiles (M136 #60 and M137 #92) exceeding 3000 m depth a stratification profile was calculated first for each profile using 20 data points with 1 dbar spacing and then a mean profile was calculated from both casts in 5 m intervals.

### 4 Results

#### 4.1 Variability of the boundary circulation

Direct velocity observations from the multi-cruise program allow a detailed description of the variability of the eastern boundary alongshore velocity structure at 12° S for a period of more than eleven weeks (Fig. 2). During the observational period from early April to June 24, 2017, we captured an event of strongly increased poleward flow that started in early May and lasted for about 35 days.

The time series started with two days of sampling in April 7 to 8. During this period, a distinct poleward flowing PCUC was present that extended 80 km offshore (Fig. 2a) and had maximum velocities of more than 0.3 m s$^{-1}$ on the continental shelf. Satellite altimetry indicated an increasing SLA towards the coast (Fig, 2a upper subpanel). Between April 18 and May 26 and except for a short break from May 4 to 6, the 12° S section was continuously occupied. In mid-April poleward flow was present only on the shelf (Fig. 2b). However, from end of April to mid-May, the PCUC considerably strengthened reaching maximum core velocities of about 0.5 m s$^{-1}$ between 50 and 100 m depth 50 km offshore (Fig. 2c, d and e). Furthermore, its poleward

flow extended to more than 80 km offshore and it occupied the whole water column above 400 m depth. The SLA increased towards the coast (by 2 cm over 40 km) implying that a poleward geostrophic velocity anomaly was present at the sea surface as well. Towards the end of May, poleward flow in the PCUC did not increase further, but remained at a similar level as during mid-May (Fig 2f). In contrary, velocity data from a final section occupation 3 weeks later in June 24 evidence that the PCUC

had weakened drastically (Fig. 2g), exhibiting a comparable velocity distribution as between April 18-25 (Fig. 2b). Assuming similar time scales for its deceleration as for it acceleration, the period of intensified PCUC flow was between 30 to 40 days. The intensified PCUC flow strongly exceeds climatological PCUC flow reported from this region. Mean alongshore flow at 12° S determined from vmADCP data sampled during 22 cruises show maximum PCUC core velocities of $0.1 - 0.15$ m s$^{-1}$ (Chaigneau et al., 2013), similar to the situation observed during April 18-25 and June 24 (Fig. 2b and g).

In April, the velocity sections indicated equatorward flow offshore and below the PCUC (Fig. 2). At these depth and offshore ranges, the Peru Coastal Current and the Chile-Peru Deep Coastal Current are thought to be located (e.g. Penven et al., 2005; Chaigneau et al., 2013). In late April, the equatorward flow increased in strength and extended to shallower depth (Fig. 2b and c). However, during the period of strong poleward flow in May, the equatorward flow decreased and was present only below 400 m depth close to the offshore end of the section (Fig. 2e and f). On June 24, weak equatorward flow was present below

200 m at most parts of the section but never reached velocities of 0.1 m s$^{-1}$. We found no indication of equatorward flow above or inshore of the PCUC, where the equatorward surface jet is expected to be situated. The lack of this surface flow in observations was also previously noted by Chaigneau et al. (2013).

To compare the alongshore circulation with hydrographic and biogeochemical sampling, the data were averaged into two periods: The initial phase of weak poleward flow (Fig. 3a) covering the period from April 18 – May 3, and a period of poleward

flow 12-26 May (Fig. 3b). The increase of poleward velocities is especially strong between 40 and 60 km offshore where velocities increase from about zero to 0.4 m s$^{-1}$ (Fig. 3c). The core of velocity increase is extending deeper than the intensified PCUC and is more detached from the coast.

## 4.2 Potential causes of circulation variability

### 4.2.1 Role of local wind stress

A potential local forcing mechanism of the intensified PCUC flow are anomalies of local wind stress curl. An increase in the magnitude of near-coastal negative wind stress curl leads to increased poleward flow along the eastern boundary through Sverdrup dynamics (e.g., Marchesiello et al., 2003). The adjustment of the circulation to changes in the wind stress curl at the eastern boundary is rather fast and occurs within a few days (Klenz et al., 2018). Wind stress curl along the Peruvian continental margin between 10° S and 14° S was negative throughout the observational period (Fig. 4), continuously forcing poleward

flow. However, during the period of PCUC acceleration between end of April and mid-May, the magnitude of negative wind stress curl decreased (Fig. 4c, d, e, f). It can thus be ruled out that local wind stress curl forcing is responsible for the observed intensified PCUC. Nevertheless, elevated negative wind stress curl was observed from May 18 – 22, which may have contributed to maintaining a strong PCUC in late-May.

Variability of near-coastal alongshore wind stress excites CTWs which propagate poleward (e.g. Yoon and Philander, 1982)

and thereby enhance or decrease poleward flow within the depth range of the PCUC. Model studies show that CTWs are excited near the equatorward edge of the region of wind variability (e.g. Fennel et al., 2012). In Mid-April through May 2017, alongshore wind stress between 6°S and 15°S was variable (Fig. 5). While moderate wind stress (0.03-0.06 N m$^{-2}$) prevailed from mid-April to May 3, it was weak during the first two weeks of May (Fig. 5d, e, g). However, during the later period the strong acceleration of the poleward flow occurred, requiring an intensification of alongshore wind stress. Thus, the initial

acceleration of the PCUC during this period (Fig. 2d, e) cannot be related to local wind stress variability. Alongshore wind stress did significantly strengthen on May 15 and remained elevated for a period of about 5 days. This wind event was intense

between 15° and 8° S, but did not occur north of 8° S. CTWs were likely excited in the region between 12° and 8° S that contributed to the elevated poleward velocities observed in the later phase between May 17 and 26 (Fig. 2f).

### 4.2.2 Equatorial winds and wave response.

A weakening of the trade winds at the equator by e.g. westerly wind events forces downwelling on the equator generates an eastward propagating equatorial Kelvin waves, which in turn may have transmitted parts of its energy to a CTW at the eastern boundary. Indeed, several westerly wind anomalies occurred in the central and eastern equatorial Pacific during the first 6 month of 2017 (Fig. 6). A particularly elevated westerly wind anomaly between the date line and 120° W occurred during the first two weeks of April (Fig. 7a). A positive SLA propagating along the equator appears to the east of the wind event at about

100° W (Fig. 7b).

This behaviour is similar to the appearance of the positive signals in the filtered SLA only east of 95° W (Fig. 6). A negative SLA anomaly occurs in the western equatorial Pacific at the time of the wave propagation and earlier (Fig. 7b) which may have been forced by the easterly wind anomaly at 160° E (Fig. 7a). But the SLA (Fig. 7b) reveals that the propagating negative SLA seen in filtered data (Fig. 6) does not occur throughout the basin and in the eastern Pacific the local minimum between

phases of higher SLA is exaggerated by the filter. A downwelling CTW causes an increase of the PCUC, an increase of SLA and downward movement of subsurface isopycnal. The SLA and PCUC change suggest the existence of a wave of this sign. A first mode downwelling CTW would induce poleward transport across the upper 1500 m and we see poleward transport throughout the measurement range in the upper 1000 m. The existence of positive of a coherent high SLA along the eastern Equator and the South-American coast with poleward propagation does support the existence of a downwelling wave generated

around 95° W as well, while the speed of the propagation suggests a first mode wave.

Similar to the velocity structure, SLA signals also support an association of the intensified poleward flow to the passage of a downwelling CTW. A downwelling CTW is associated with an upward elevation of the sea surface and a compensating downward displacement of the isopycnals in the water column as well as an intensification of poleward flow (e.g. Echevin et

al., 2014). In this study we use the designation "downwelling" only to indicate the sign of velocity and SLA anomalies associated with the CTW. As discussed in the previous section, a local SLA increase was observed at the Peruvian coast while elevated poleward velocities within the PCUC depth range were present.

Bandpass-filtered SLA data from near the continental slope (section 3.3) indicates a positive SLA off Peru and Ecuador between the equator and about 14° S during this period (Fig. 6). The positive SLA along the coast propagates poleward at a

velocity not inconsistent with a propagation speed of 3.1 ms$^{-1}$, the phase speed of the first CTW mode (Fig. 5). Moreover, when looking at SLA along the equator, there is a coherent signal starting at about 95° W moving towards the eastern boundary and arriving at about the same time when the SLA maximum of the coast is developing. Again, signal propagation from west to east is indicated, which agrees with the phased speed of a first vertical mode equatorial Kelvin waves (e.g. Yu and McPhaden, 1999). The SLA indicates the propagation of negative anomalies corresponding to an upwelling wave about 20

days earlier and with origin west of 140° W (Fig. 6) the arrival of this potential upwelling wave fits with the downward tilt in the SLA between April 25 and May 3 (Fig. 2c) and may contribute to the weak poleward flow by causing equatorward velocity anomalies.

### 4.1.3 Modal structure of the intensified flow

The cross-shore-depth structure of alongshore velocity obtained for the first three CTW modes at the 12° S section (see section

3.4) varies predominately in the vertical axis with poles of opposing velocity located above each other (Fig. 7). Flow reversal for each individual mode occurs at shallower depth away from the boundary compared to inshore regions. As expected, higher modes exhibit an upper pole of enhanced velocity at shallower depth compared to lower modes and the phase speed (Fig. 7)

decreases. The obtained phase speeds are within ranges reported by Illig et al. (2018a, their table 1) for the first two modes while the phase speed of the third mode is slightly lower (0.82 ms$^{-1}$ compared to 0.93±0.08 ms$^{-1}$). The velocity structure of the modes is very similar to their structure reported in their study at 16° S as well.

For comparison, we show the difference of full-depth OS38 vmADCP alongshore velocities between May 12 – 26 and April 18 – May 3 (Fig. 7e). Apart from the increased poleward velocity in the upper 500 m, the alongshore velocity difference is weakly poleward throughout the upper 1000 m of the water column resolved by the OS38 (Fig. 7e). When comparing the baroclinic structure of the observations to the baroclinic structure of the different CTW modes, it becomes obvious that due to the missing flow reversal, the observed change in alongshore velocity is best described by a first mode CTW. This mode features poleward flow anomalies throughout the upper 1500 m (Fig. 7b) which agrees with the distribution of the velocity differences between the two time periods throughout most of the upper 1000 m (Fig. 7e). The maximum increase of poleward flow, on the other hand, is restricted to a depth range similar to the upper poleward velocity core of a second mode CTW.

## 4.3 Response of hydrographic conditions to the PCUC intensification

In the following we analyse the changes in hydrographic conditions co-occurring with the increase of alongshore flow. Lower near-surface conservative temperatures near the coast compared to offshore (Fig. 1, Fig. 8a, b) indicated active upwelling during the observational program. While the upwelling signal was restricted to the upper 50 m, near-coastal water masses between 50 m and 300 m were significantly warmer compared to water masses offshore (Fig. 8a). During the intensified PCUC period (Fig. 8 middle panels) the cross-shore temperature gradient intensified, leading to an increased downward displacement of isopycnals and isotherms near the coast (Fig. 8b). There, the associated warming signal between 100m and 200 m depth was up to 0.5° C. It note that during the PCUC intensified period, near-coastal surface temperatures decreased.

Absolute salinity featured a shallow subsurface salinity maximum at about 25 m depth originating offshore and extending over the slope and shelf (Fig. 8d and e). As for temperature, cross-shore salinity gradients were evident between 50 and 300m depth with higher salinities near the coast which intensified when the PCUC increased. At the same time, salinity at the upper shelf decreased (Fig. 8f).

Distribution of oxygen concentrations were characterized by a sharp oxycline above the anoxic OMZ (Fig. 8g and h). During the weak PCUC period, oxygen concentrations decreased from slightly supersaturated concentrations at the surface to anoxia within the upper 100 m of the water column (Fig. 8g). At depth between 450 and 500 m, oxygen concentrations started to increase again to detectable values (Fig. 8g). When the poleward flow intensified, low oxygen waters were found deeper in the water column following the downward displacement of the isopycnals (Fig. 8h). During this period, oxygen concentrations of 2 µmol kg$^{-1}$ were found at 200 m depth in the bottom water (Fig. 8h), which has significant consequences for benthic and pelagic biogeochemical processes in that depth range discussed below.

In the upper water column above 400m, waters denser than 1025.9 kg m$^{-3}$ were mainly Equatorial Subsurface Water (ESSW; Fig. 9). ESSW originates from the equatorial current system. It is characterized by a linear relationship of temperature and salinity in the temperature range 8 – 14 °C and absolute salinity range 34.6 – 35.0 (e.g. Grados et al., 2018). Lower salinity Eastern South Pacific Intermediate Water (temperature range 12 – 14 °C, salinity 34.8), which is also situated in the depth range mentioned above, was only observed in the hydrographic data from two offshore stations (Fig. 9a). The dominance of ESSW was not affected by the increasing poleward velocities (Fig. 9b) and most profiles follow the same temperature and salinity relationship in both phases. In fact, during PCUC intensification, the ESSW was the sole water mass in the upper 400m within 80 km of the coast (Fig. 9b).

## 4.4 Response of nutrient biogeochemistry to the PCUC intensification

Nutrients are transported poleward by the PCUC and their advection thus is likely influenced by the variability of poleward velocity. In the following we describe the observed changes in nutrient concentrations and relate them to the variability in PCUC strength.

Nitrate concentrations on the shelf and upper slope increased when the poleward flow strengthened (Fig. 10a and b). The nitrate concentrations were low at the surface and increase with depth (Fig. 10a and b). During the initial phase, offshore surface nitrate concentrations decreased to less than 10 µmol kg$^{-1}$ and increased below to 20 µmol kg$^{-1}$ at 50 m and 25 µmol kg$^{-1}$ at 300 m depth (Fig. 10a). Low concentrations in bottom waters on the shelf were most prominent between 75 and 100 m depth going down to 15 µmol kg$^{-1}$ (Fig. 10a). After the intensification of the PCUC, nitrate concentrations between 50 and 100 m depth offshore and 250 m in bottom waters increased and exceed 25 µmol kg$^{-1}$ (Fig. 10b). Throughout this part of the section the increase exceeded 2.5 µmol kg$^{-1}$ (Fig. 10c), including areas with an increase in excess of 5 µmol kg$^{-1}$ and even up to 10 µmol kg$^{-1}$. The surface layer featured anincrease in excess of 5 µmol kg$^{-1}$ as well (Fig. 10c).

Nitrite concentrations in the bottom water on the shelf and upper slope were reduced by the intensified PCUC (Fig. 10d and e). The nitrite concentrations were low outside of the OMZ and their structure featured two maxima, the main one located in the centre of the OMZ around 300 m depth and the secondary maximum in the upper part of the OMZ at 150 to 200 m depth. After the intensification of the PCUC the upper boundary of nitrite containing water was displaced downwards, leaving bottom waters above 250 m depth free of nitrite (Fig. 10e). The depletion of nitrite in the bottom water was coupled to a weak ventilation supplying oxygen (Fig. 8h). This caused a nitrite decrease exceeding 2 µmol kg$^{-1}$ in the bottom water around 200 m depth (Fig. 10f).

Ammonium concentrations were generally low or undetectable. Concentrations in excess of 0.4 µmol kg$^{-1}$ occurred only on the upper shelf and close to the surface (Fig. 10g and h), and were indicative of remineralisation of phytoplankton detritus, with rapid removal over time of the ammonium due to phytoplankton uptake and nitrification. The patchiness of ammonium concentrations caused high positive and negative differences very close to each other on the shelf (Fig. 10i). In the surface layer above 50 to 80 m in offshore waters, a decline of ammonium concentrations was observed.

Phosphate concentrations did not change strongly by the increased strength of the PCUC (Fig. 10j and k). Concentrations were low at the surface and increased to 2 µmol kg$^{-1}$ at 50 m depth. Within the upper OMZ the concentrations were higher offshore than onshore (Fig. 10j and k). When the PCUC intensified, concentrations decreased below 50 m depth and above 100 m at 80 km offshore and 300 m inshore (Fig. 10l). A phosphate decrease of up to 0.3 µmol kg$^{-1}$ occurred in bottom waters on the shelf at water depths shallower than 100 m.

The nitrogen deficit was reduced in the later phase of the strong PCUC (Fig. 10m and n). During the weak PCUC phase the offshore maximum deficit was located between 150 and 200 m depth exceeding 12.5 µmol kg$^{-1}$ and the absolute maximum was a localized peak exceeding 15 µmol kg$^{-1}$ in bottom waters just above 100 m depth (Fig. 10m). This maximum on the shelf was caused by low nitrate (Fig. 10a and high phosphate concentrations (Fig. 10j), while nitrite (Fig. 10d) and ammonium (Fig. 10g were enhanced as well. After the PCUC intensification the deficit reached 5 µmol kg$^{-1}$ at about 70 m depth offshore and 200 m depth at the coast (Fig. 10n). At greater depths the maximum deficit exceeding 12.5 µmol kg$^{-1}$ was located offshore around 150 m depth, extending towards the coast along isopycnal surfaces. After the PCUC intensification, the nitrogen deficit in the upper 200 m inshore of 70 km was reduced (Fig. 10o). The maximum decrease of the deficit occurred in the bottom water just above 100 m depth exceeding 10 µmol kg$^{-1}$; here the maximum described above in Figure 7(m) disappeared with the PCUC intensification. Further offshore around 100 m depth the decrease of the deficits exceeded 5 µmol kg$^{-1}$ in several patches as well (Fig. 10o).

The decrease in nitrogen deficit between 50 m depth and about 200 m depth (deeper toward the coast, and shallow offshore) exceeding 5 µmol kg$^{-1}$ in its maximum agreed with the location of largest poleward flow (Fig. 3b). At this vertical and horizontal range nitrate was increasing by more than 2.5 µmol kg$^{-1}$ (Fig. 10c) and phosphate decreased (Fig. 10l), both changes

contributed to the reduced nitrogen deficit. Nitrite concentrations decreased as well (Fig. 10f), but the total increase in nitrogen

species still exceeded the phosphate decrease by more than the ratio implied in equation (1), with the nitrate change, dominating the change of nitrogen species.

The increase in nitrate concentrations and the decrease of the nitrogen deficit and phosphate concentrations are strongest in the upper 200 m where the intensified PCUC had its maximum during the same time period. These changes occurred when comparing both regimes in density space as well (not shown) and a nitrate increase within the ESSW was evident This suggests

that poleward advection in the intensified PCUC is the main cause for the changes in biogeochemical properties while changes due to vertical displacement only play a minor role, here.

## 5 Summary and Discussion

Measurements from an intensive physical and biogeochemical shipboard sampling program off Peru at 12° S are used to analyse intraseasonal variability of the eastern boundary circulation and associated changes in hydrography and nutrient

distributions. The most prominent finding is an intensification of poleward velocities within the depth range occupied by the PCUC that occurred throughout the last 3 weeks of May in 2017. During this period, maximum poleward velocities in the PCUC core between 50 m and 100 m depth were above 0.5 m s$^{-1}$ exceeding the reported mean state by far (e.g. Chaigneau et al., 2013). The poleward flow occupied the whole water column above 1000 m depth and extended to more than 80 km offshore.

The elevated poleward velocities at the eastern boundary were likely associated with a passing downwelling CTW. Satellite SLA data indicated a poleward propagation of a positive SLA signal from the equator to beyond 14° S that occurred simultaneously to the increase of poleward flow at 12° S. The SLA signal propagated at a speed consistent with the phase speed of the first vertical mode CTW. Similarly, the vertical distribution of the poleward velocity anomaly at the boundary was consistent with the cross-shore-depth velocity structure of a first vertical mode CTW.

Previous studies have identified the first vertical mode CTWs to dominate intraseasonal variability in the eastern South Pacific based on observations (Brink, 1982; Shaffer et al., 1997) and model results (Illig et al., 2018b). However, observed intraseasonal intensification of poleward flow within the depth range of the PCUC in a previous study by Pietri et al. (2014) was attributed to the second and third mode CTW modes. They found poleward velocities along the Peruvian continental slope increasing to 0.4 m s$^{-1}$.

Although the observed cross-shore-depth velocity structure of the CTW generally agrees with the first vertical mode solution of a linear wave model using local stratification and topography, there is disagreement in the details of the flow structure. It was noted previously that observed velocity structures agree poorly (Brink, 1982). Additionally, we compare only the theoretical modal structures with the difference between two velocity phases and the initial phase is no solid estimate of the mean alongshore circulation off Peru. Another possible explanation of the poor agreement between the first mode CTW and

the observed flow intensification is interaction of the CTW with local topography. North of our sampling site, the continental slope bends offshore at depths between 500 m to 1000 m (Fig. 1, insert) while the shelf narrows to the south. Changes in coastline, shelf width, and along-slope bathymetry lead to a transfer of CTW energy into higher modes (scattering) and upstream backscattering (Wang, 1980; Wilkin and Chapman, 1990; Kämpf (2018); Brunner et al., 2019). The influence of changes in shelf width on the upstream alongshore flow structure can extend to 200 km upstream (Wilkin and Chapman, 1990)

and is likely not relevant here. However, the bent of the continental slope north of out sampling site may stimulate energy transfer into high vertical mode CTWs. In turn, the superposition of several vertical modes could explain the observed elevated poleward flow between 50 m and 300 m depth. In fact, a recent model study suggests that differences between the theoretical CTW solutions and observations are predominately due to wave scattering (Brunner et al., 2019).

Furthermore, a recent study suggests that differences between the theoretical CTW solutions and observations are due to wave
scattering into a greater proportion of higher-order modes or backscattering upstream (Brunner et al., 2019). Scattering occurs
when a propagating CTW encounters changes in coastline, shelf width, and bathymetry. These changes were neglected when
deriving the CTW mode solutions used here.

The SLA data indicate that the first vertical CTW mode along the eastern boundary that arrived at our sampling site in mid-
May originated in the eastern equatorial Pacific at around 95° W (Fig. 6). A coherent signal from this region propagated
eastwards along the equator and arrived at about the same time when the equatorial SLA maximum at the coast was developing.
The wind data shows a possible forcing of this wave around 160° W in early April by a weakening of zonal wind (Fig. 7a).
This is consistent with a recent comprehensive study (Rydbeck et al., 2019) identifying wind forcing variability west of 150°
W as the main generation mechanism of intraseasonal equatorial Kelvin waves.

However, in the SLA the wave has an expression on east of 95° W compared to 150° W where Rydbeck et al. (2019) have
located the maximum of SLA signal. The location of the SLA signal to the eastern equatorial Pacific may have been caused
by the negative SLA further in the west which reduces the positive SLA to an extent where it is no longer identifiable.

The temperature and salinity conditions on the shelf remain almost unchanged despite the strongly intensified poleward flow
and are only displaced downwards, suggesting that the same water mass was advected within the boundary current regime
during both observational periods. The weak warming and increase in salinity agree with the advection of the slightly warmer
and saltier water along the PCUC path north of 12° S (Grados et al., 2018). However, the signals are rather weak therefore we
cannot relate them clearly to the circulation change. The SST declines despite the downwelling wave which would be expected
to cause warming. A downwelling CTW in March 2017 indeed has contributed to the warm SST anomalies (Echevin et al.,
2018). However, the impact of CTWs on intraseasonal SST variability off Peru is limited in general (Dewitte et al., 2011; Illig
et al., 2014). Specifically, in May 2017 SST reduction agrees with both the seasonal cycle (Graco et al., 2017) and the decline
of the warm Coastal El Niño with peak SST in March (e.g. Garreaud, 2018).

Despite being non-conservative, the changes in oxygen concentrations are small as well. Only in the bottom water on the upper
continental slope, higher oxygen concentrations are displaced downwards. Faster advection should cause higher oxygen
concentrations, because the time is shorter, in which oxygen is consumed by respiration along the PCUC path. Espinoza-
Morriberón et al. (2019) attribute some of the oxygen variability associated with El Niño-Southern Oscillation to the changes
in microbial respiration because advection speed and pathways change under El Niño conditions (e.g. Montes et al., 2011;
Espinoza-Morriberón et al., 2017). However, Zamora et al. (2012) reported uniform oxygen concentrations for waters in the
PCUC; therefore, there is no oxygen change due to changes in alongshore advection, explaining the weak oxygen change in
our study.

The limited impact of the CTW on oxygen may be related to high oxygen concentrations present at the study site before the
wave passage. Compared to time series data of Callao (Graco et al., 2017) the concentrations were in the upper range of
variability observed after the 1997-98 El Niño event. The 22 µmol kg$^{-1}$ surface was located at 50 m instead of 30 m reported
for the climatological state by Espinoza-Morriberón et al. (2019). Whether the higher oxygen concentrations are caused by the
coastal El Niño event, in the same way that canonical El Niños are related to enhanced oxygenation (e.g. Helly and Levin,
2004; Stramma et al., 2016; Espinoza-Morriberón et al., 2019), cannot be answered with our available observational data.
Coastal time series off northern Peru show an oxygenation starting in February 2017 but only a weaker signal at 12° S (ENFEN,
2017) while hydrographic profiles collected in early to mid-April at 11 and 14° S feature a very shallow oxycline (not shown).
Therefore, it remains unclear whether the higher oxygen concentrations in late-April and May at 12° S are a local or short-
term phenomenon or indeed related to the coastal El Niño event.

The increase in nitrate concentrations and the reduced nitrogen deficit are likely caused by the shorter advection timescales in
the intensified flow. The nitrate increase occurs within the ESSW range and waters with the same T-S properties are richer in
nitrate after the PCUC increase (Fig. 9), excluding changed advection pathways as a likely cause of the increased nitrate load.

Along the ESSW pathway within the PCUC from the equator to 12° S, the nitrogen deficit increases while nitrate concentrations decrease (Silva et al., 2009; Zamora et al., 2012; Kalvelage et al., 2013).. The increasing nitrogen deficit is caused by the microbially facilitated reduction of nitrate, nitrite and ammonium to $N_2$ gas which occurs in anoxic waters during the consumption of organic matter (e.g. Kalvelage et al., 2013). The resulting nitrogen deficit accumulates with time during the poleward advection. Thus, the intensified PCUC increases the poleward advection of water with a lower nitrogen deficit. The possibility of this mechanism is tested by calculating the advection timescales from the equatorial regime: the 12° S section is about 1800 km alongshore distance away from the equator, the advection timescale for a velocity of 0.4 m s$^{-1}$ is 52 days, compared to 160 days for a velocity of 0.13 m s$^{-1}$, approximately the climatological PCUC velocity (Chaigneau et al., 2013). Using an N-loss of 48 nmol N l$^{-1}$d$^{-1}$ (combined anammox and denitrification rates in the coastal OMZ from Kalvelage et al. (2013)), 5.2 µmol N l$^{-1}$ can be transformed during the longer advection timescale. This may explain the reduction of the nitrogen deficit by about 5 µmol N kg$^{-1}$, as observed throughout much of the PCUC core. The sediments off Peru below the OMZ release phosphate into the water column (Noffke et al., 2012; Lomnitz et al., 2016) that also contributes to the nitrogen deficit. Shorter advection timescales lead to a reduced accumulation of benthic phosphate release and in fact phosphate concentrations decrease during the strong PCUC flow. However, the increase of nitrate (and the sum of inorganic nitrogen species) exceeds the phosphate decrease by a ratio higher than the nitrogen to phosphorus relation implied in equation 1. Therefore, changes of nitrate concentration dominate the reduction of the nitrogen deficit.

The increase of nitrate by the downwelling CTW implies that the change of alongshore advection with the increased flow is more important for the nitrate balance than downwelling. The downwelling would displace the nutricline and thus low nutrient surface water downwards, lowering nutrient concentrations, which is not observed. The decline of nitrate concentrations during El Niño events has been associated with the nutricline displacement due to downwelling CTWs on interannual timescales (Graco et al., 2017; Espinoza-Morriberón et al., 2017). However, focusing on intraseasonal timescales, Echevin et al. (2014) modelled an almost cancelling of horizontal and vertical (i.e. nutricline movement) advection and a fast mode CTW not impacting nutricline depth. In a model study in the Atlantic Ocean, where nitrate decreases poleward as well, it was shown that the total effect of CTWs on nitrate concentrations varies regionally due to a different balance of horizontal and vertical advection (Bachèlery et al., 2016), but the horizontal advection always led to an increase in nitrate.

The changes in redox state in the water column and especially the bottom water affect the biogeochemical cycling in the sediment as well. Because microbial storage of nitrate and nitrite by microorganisms in the sediment can sustain vigorous N turnover even in the absence of bottom water nitrate and nitrite (Dale et al., 2016; Sommer et al., 2016), episodic events of nitrogen supply can be associated with continuous benthic nitrogen cycling. The absence of nitrate supply due to the absence of CTWs over longer time periods favours the depletion of nitrate in the water column as observed by Sommer et al. (2016) and may lead ultimately to the development of sulfidic events (Schunck et al., 2013; Dale et al., 2017; Callbeck et al., 2018).

## 6 Conclusion

Based on extensive physical and biogeochemical sampling, we describe and analyse the evolution of circulation, hydrography and biogeochemistry off Peru in early 2017. Poleward velocities within the PCUC intensified far beyond the reported climatological mean in May. This increased flow occurs during a limited time period in May in between weaker poleward transport in April and late June. The propagation velocity of positive SLA along the equator and coastline suggests that the intensified current is caused by a poleward propagating downwelling CTW of the first baroclinic mode forced around 160° W at the equator that causes a positive SLA signal east of 95° W. The transition of the circulation from a weak poleward flow to strong poleward flow decreased the timescale of alongshore advection from the equatorial current regime to the study site at 12° S.

The downwelling CTW is not associated with strong vertical displacements of waters; instead the advection caused by the intensified PCUC is more important. For parameters without strong horizontal gradients, an increase in PCUC flow does not cause pronounced changes in the advection. In this study this applies to the conservative properties temperature and salinity as well as for oxygen where alongshore gradients are weak (Zamora et al., 2012). For these parameters there are no large differences between both circulation phases that can be attributed clearly to the altered circulation. Yet concentrations of nutrients are influenced by shorter transit times, being less altered by biogeochemical cycling. Thisleads to an increase of bioavailable nitrogen in the OMZ. and its biogeochemistry is strongly changed by the increased ratio of nitrogen to phosphorus related to the increased advection..

For the period from April to May 2017, our study suggests an increase in nitrate levels due to the passage of an intraseasonal downwelling CTW. This contrasts with the decrease observed previously on interannual timescales caused by downwelling CTWs (Graco et al., 2017). This shows that the impact of CTWs on nutrient biogeochemistry is a complex balance between different factors, with potentially different outcome on different timescales. Analysing the processes associated with individual intraseasonal waves is also necessary to understand the interannual effect of CTWs, which is based on varying occurrence of such waves in different years.

The high variability of circulation, nutrients and the nitrogen deficit demonstrates the need for temporally resolved sampling as an individual section recorded may be very different from the situation a few weeks later. Quantification of intraseasonal variability in CTWs and their impact is – other than in modelling studies (e.g. Echevin et al., 2014) – only possible by sampling at high temporal resolution.

**Data availability**

Ship based observations are available at PANGAEA (https://doi.pangaea.de/10.1594/PANGAEA.903828). Global Ocean Gridded L4 sea surface heights are made available by E.U. Copernicus Marine Service (CMEMS). NOAA ERSSTv5 data are made available by the NOAA National Centers for Environmental Information. ASCAT data were obtained from the Centre de Recherche et d'Exploitation Satellitaire (CERSAT), at IFREMER, Plouzané (France).

**Author contributions**

JL carried out the data analysis and wrote the main manuscript. JL conceived the study with MD and with input by DC, ST and MV. MD and SS lead the observational program at sea. GK calibrated and processed the CTD. AD and SS provided the nutrient data. EA provided the ammonium data. All co-authors reviewed the manuscript and contributed to the scientific interpretation and discussion.

**Competing Interests**

The authors declare that they have no conflict of interest.

**Acknowledgements**

This study was funded by the Deutsche Forschungsgemeinschaft as part of the Sonderforschungsbereich 754 "Climate–Biogeochemistry Interactions in the Tropical Ocean". ST received funding by the European Commission (Horizon 2020 programme, MSCA-IF-2016, proposal number WACO 749699: Fine-scale Physics, Biogeochemistry and Climate Change in the West African Coastal Ocean). We thank the Peruvian authorities for the permission to carry out scientific work in their national waters. We thank the captains and the crew of R/V Meteor for their support during the cruises. We thank Regina

Surberg and Bettina Domeyer for the nutrient analysis, all other people involved in the measurement program and Rena Czeschel for post-cruise processing of the vmADCP data. We thank two anonymous reviewers for their helpful suggestions.

Many figures in this study use colour maps from the cmocean package (Thyng et al., 2016). Global Ocean Gridded L4 sea surface heights were made available by E.U. Copernicus Marine Service (CMEMS).

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

**Table 1: Dates of R/V Meteor cruises conducted within the ETSP in 2017 including sampling time and number of CTD stations collected 12° S.**

| R/V Meteor cruise | Dates in 2017 | Sampling duration along 12° S | CTD/nutrient stations at 12° S |
|---|---|---|---|
| M135 | March 2 – April 8 | 2 days | |
| M136 | April 18 – May 3 | 15 days | 59 profiles |
| M137 | May 6 – 29 | 23 days | 92 profiles |
| M138 | June 1 – July 5 | 1 day | |

800

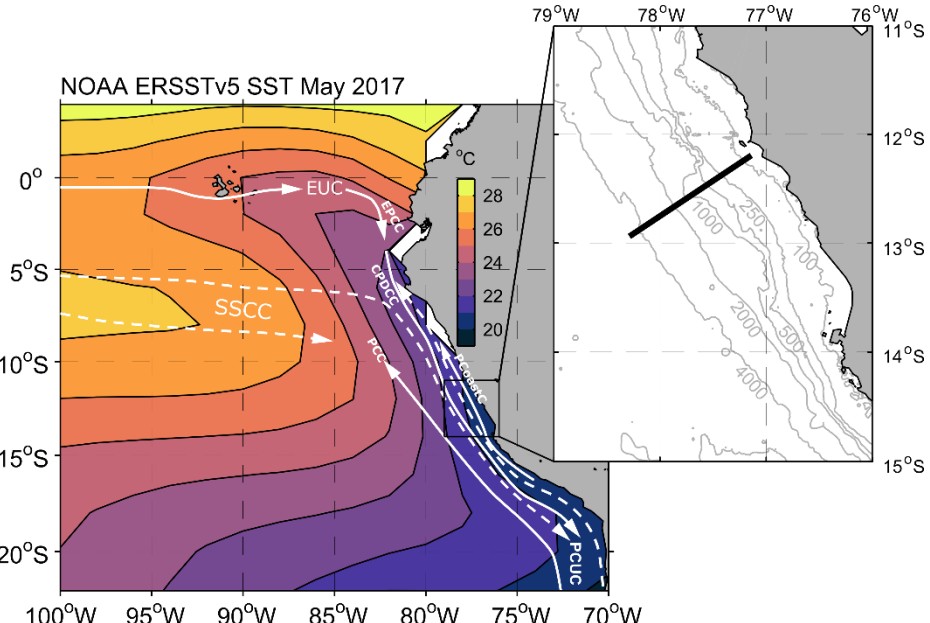

**Figure 1: Sea surface temperature (NOAA ERSSTv5 SST) in the eastern tropical Pacific during May 2017 and schematic of the circulation. White solid lines indicate surface layer currents while dashed lines indicate thermocline layer currents (after Brandt et al., 2015). Abbreviated currents are the Equatorial Undercurrent (EUC), the Ecuador-Peru Coastal Current (EPCC), the Peru Coastal Current (PCoastC), the Peru Chile Current (PCC), the Southern Subsurface Countercurrents (SSCC, with two branches), the Chile–Peru Deep Coastal Current (CPDCC), and the Peru–Chile Undercurrent (PCUC). Figure insert shows the position of the 12° S section and local bathymetry (SRTM30 plus topography, Becker et al., 2009).**

815

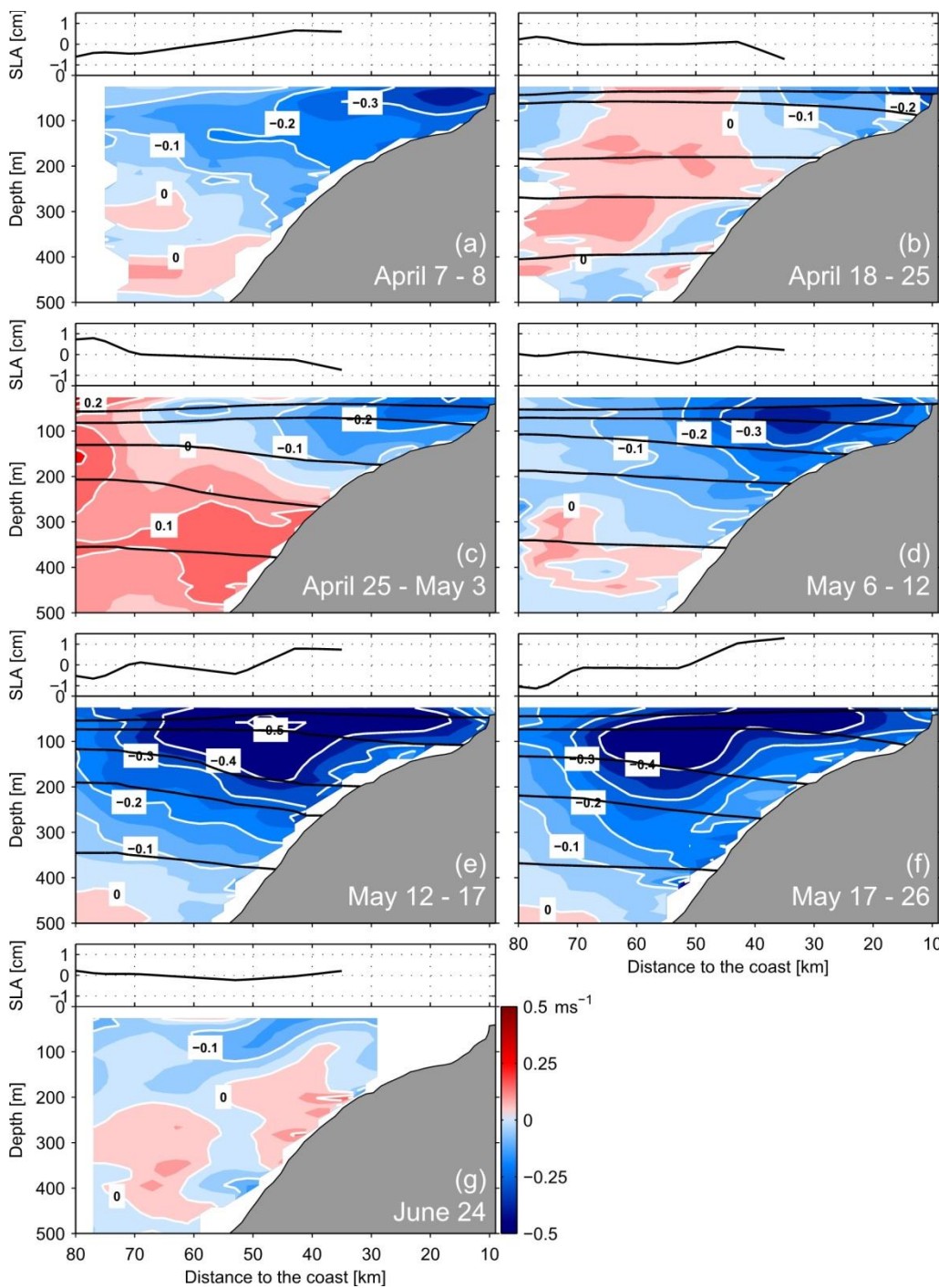

**Figure 2: Depth-distance distribution of alongshore velocity (in m s⁻¹, lower subpanels) and sea level anomaly (in cm, upper subpanels) at 12° S during April 7 – 8 (a), April 18 –25 (b), April 25 – May 3 (c), May 6 -12 (d), May 12 – 17 (e), May 17 – 26 (f) and June 24, 2017 (g). Black lines indicate the distribution of isopycnals (σ₀) 25.6, 25.9, 26.2, 26.4, and 26.7 kg m⁻³.**

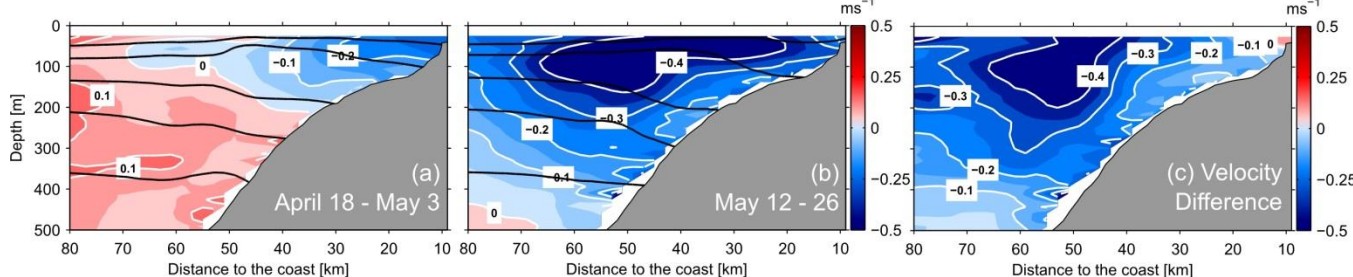

 **Figure 3: Depth - distance distribution of averaged alongshore velocity at 12° S prior to (a, April 18 – May 3, 2017) and during (b, May 12 – 26, 2017) the PCUC intensification period. Panel c depicts the velocity difference between the two situations. Black lines indicate the distribution of isopycnals ($\sigma_\theta$) 25.6, 25.9, 26.2, 26.4, and 26.7 kg m$^{-3}$.**

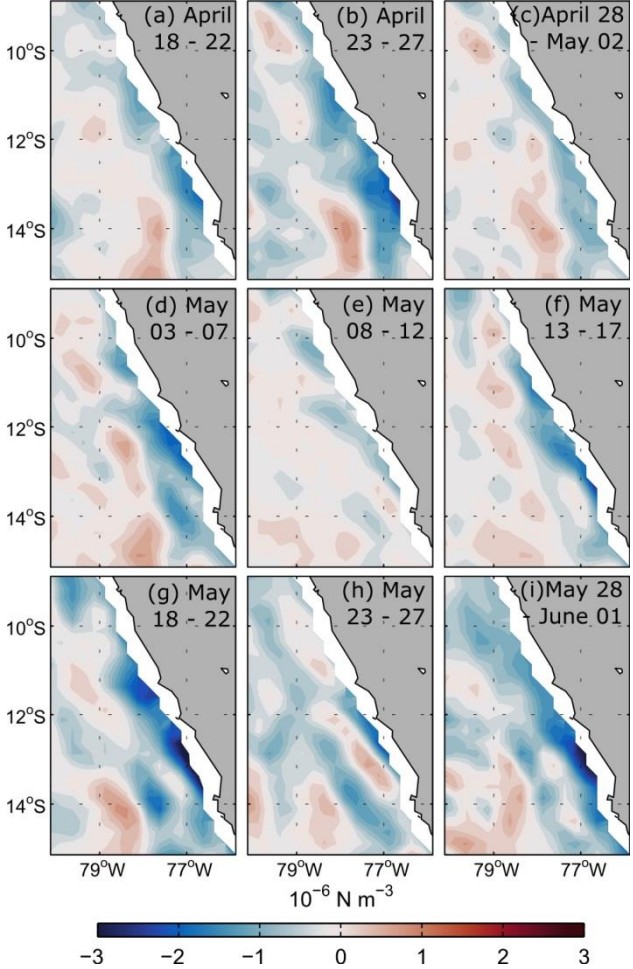

**Figure 4: Five-day mean wind stress curl from ASCAT scatterometer winds off Peru during April 18 – 22 (a), April (23 – 27 (b), April 28 – May 02 (c), May 03 – 07 (d), May 08 – 12 (e), May 13 – 17 (f), May 18 – 22 (g), May 23 – 27 (h) and May 28 – June 01, 2017 (i).**

840

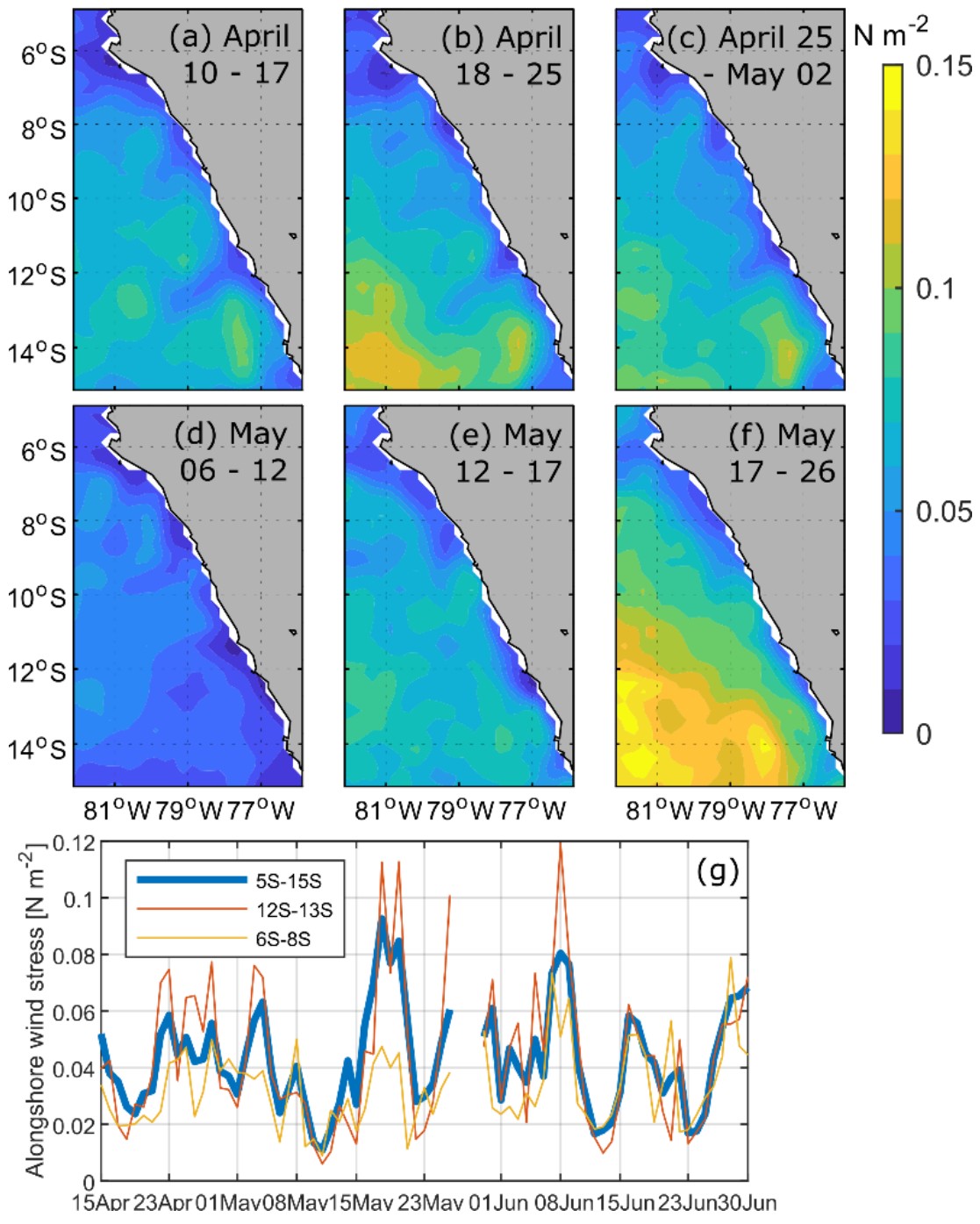

845

**Figure 5: Weekly averages of alongshore wind stress from ASCAT scatterometer winds (upper two panel rows) for April 10 – 17 (a), April 18 -25 (b), April 25 – May 2 (c), May 06 – 12 (d), May 12 – 17 (e), May 17 – 26 (f). Lower panel (g) shows time series of alongshore wind stress averaged between 30 and 80 km from the coast for different bands of latitude.**

850

855

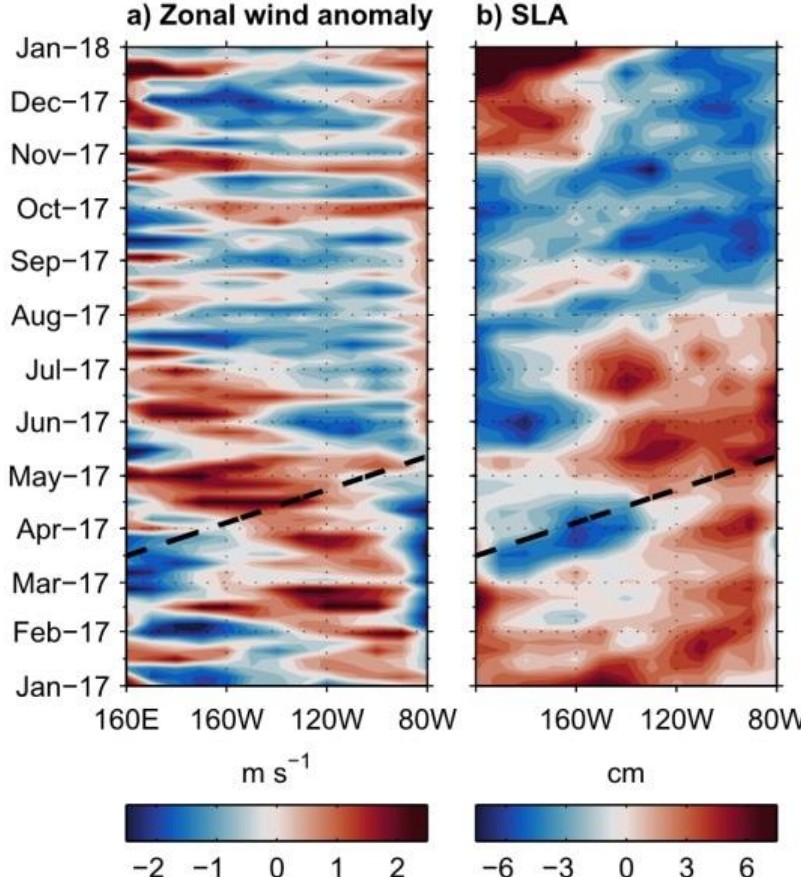

**Figure 6: Hovmöller diagram of zonal wind anomaly (left panel) and SLA (right panel) along the central and eastern equatorial Pacific for the year 2017. The propagation of a first mode equatorial Kelvin wave is indicated by the dashed black line (phase speed 2.7 m s⁻¹; Yu and McPhaden, 1999).**

860

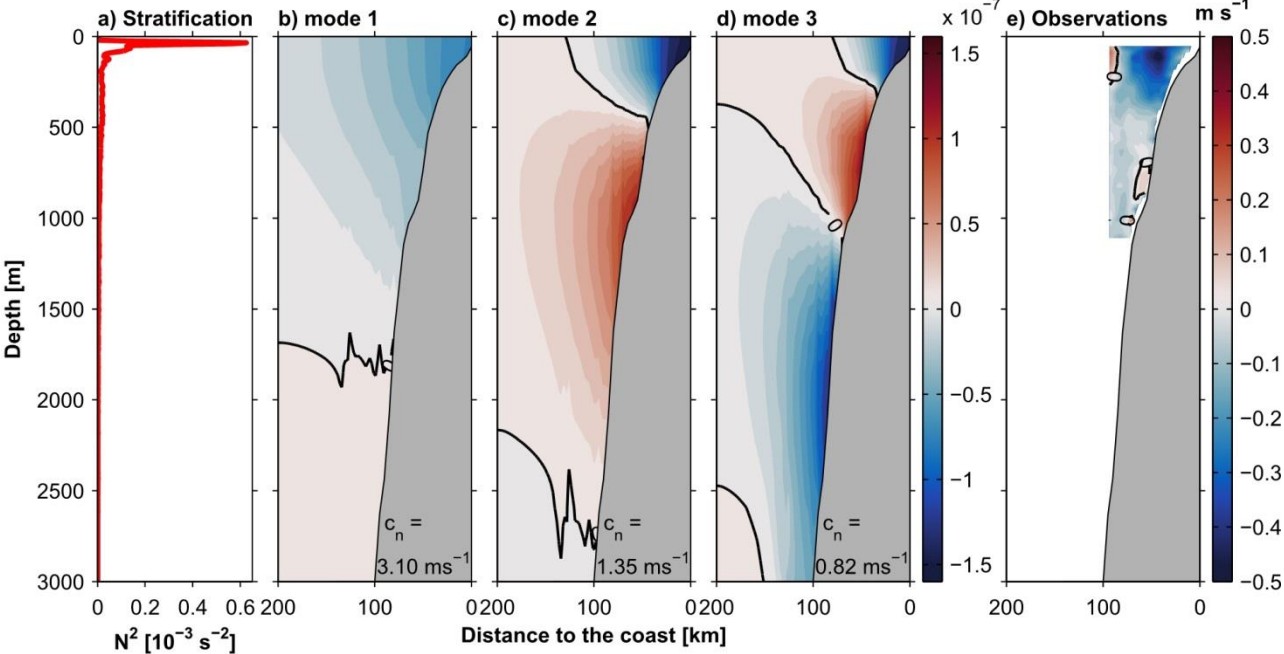

**Figure 7:** Profile of stratification (N²) (left panel) and cross-shore-depth structure of alongshore velocity (arbitrary amplitude) obtained for the first three CTW modes (panel b through d). Panel e) shows the difference of observed alongshore velocity between April 18 – May 03 and May 12 – 26, 2017. Note that the CTW phase speeds $c_n$ are given in the lower right corner of the respective velocity structure panel.

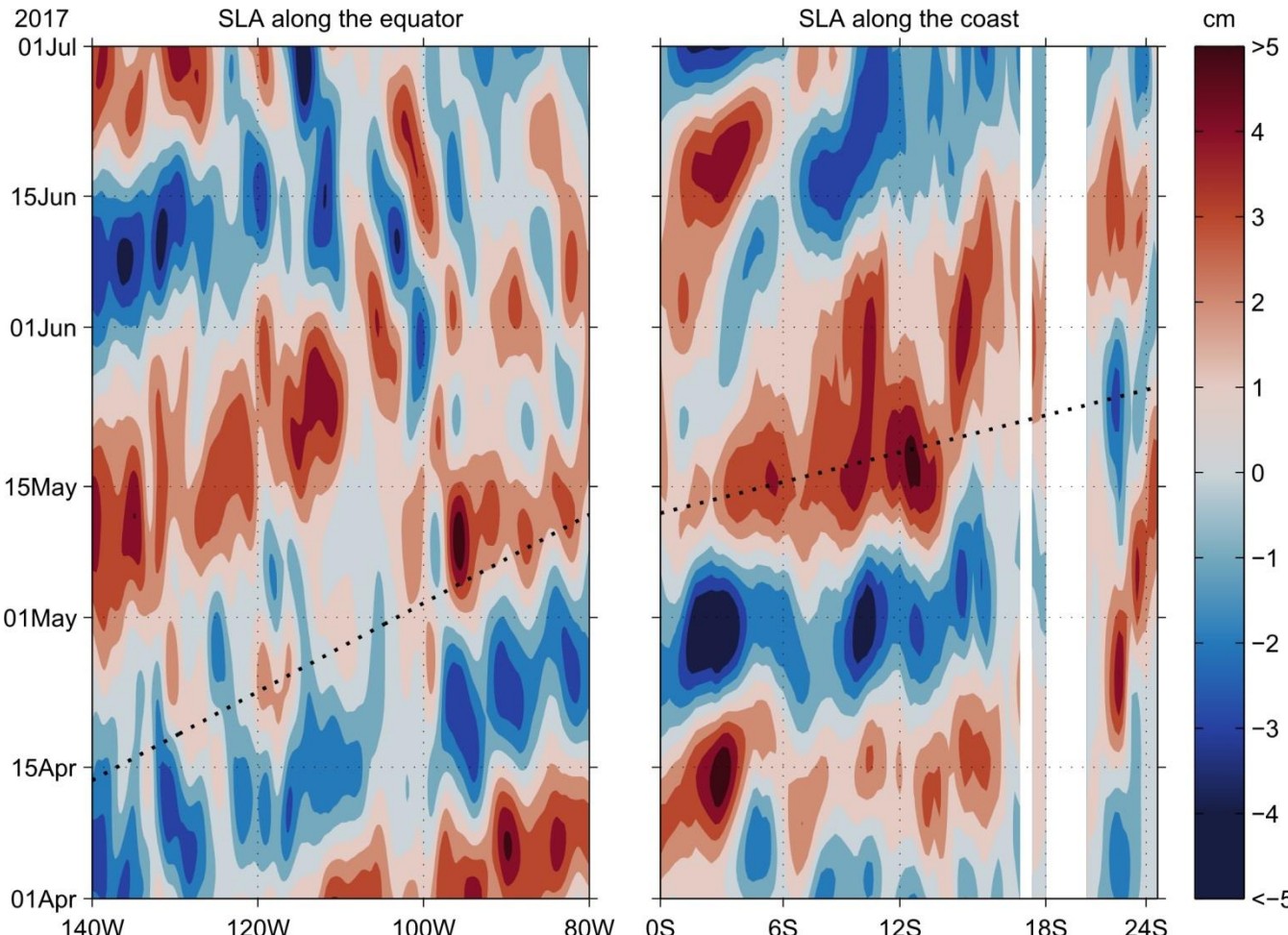

**Figure 8: Bandpass filtered (20 – 90 days) sea level anomaly along the eastern equatorial Pacific averaged between 0.25°N and 0.25°S (left panel) and along the western coast of South America (right panel) averaged over the two grid point closest to the coastline. The propagation of a first mode equatorial Kelvin wave and CTW are shown as dotted black lines, phase speed of the equatorial wave is 2.7 m s⁻¹ (Yu and McPhaden, 1999) and 3.1 m s⁻¹ for CTWs (see figure 7b).**

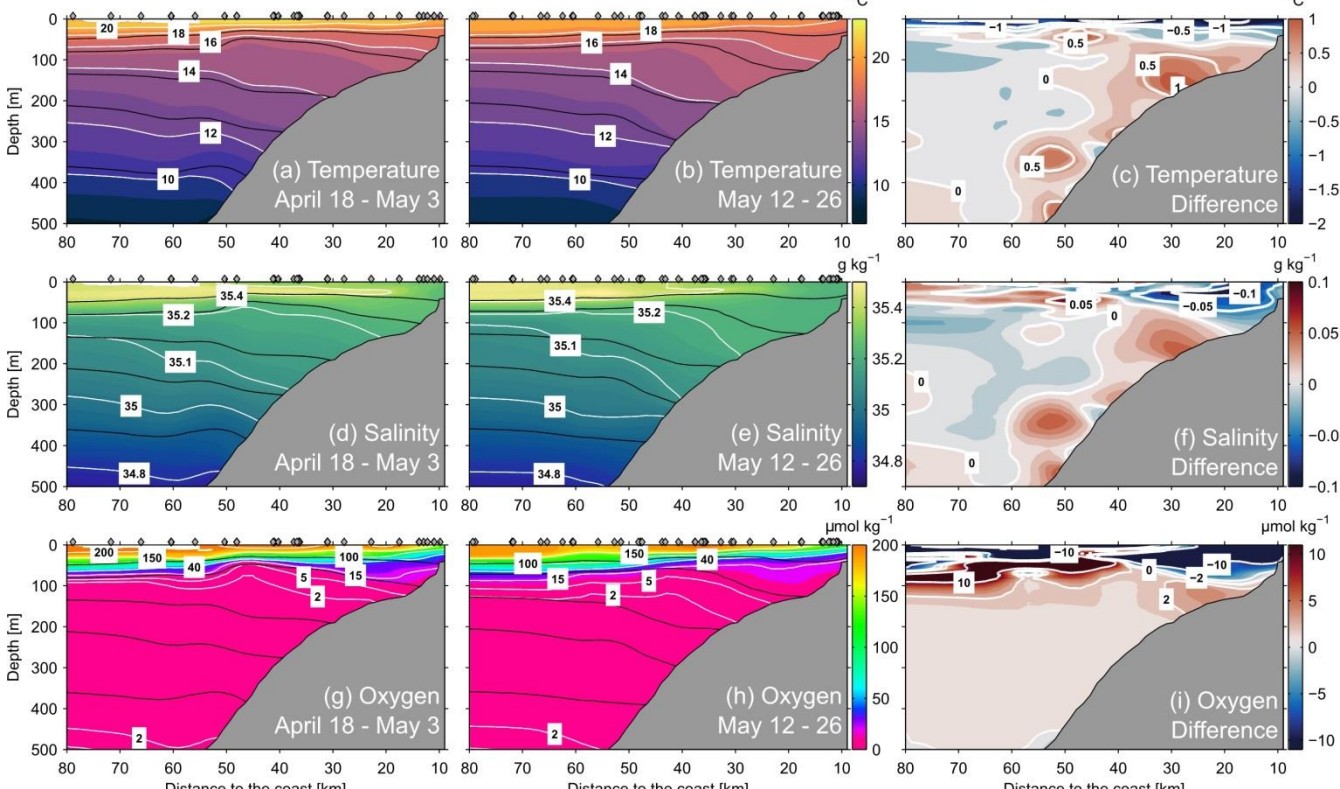

885 **Figure 9: Conservative Temperature (upper panel row), absolute salinity (middle panel row), and oxygen (lower panel row) at 12° S during April 18 – May 3, 2017 (left panel column) and May 12 – 26, 2017 (middle panel column). The difference of the respective characteristic between the two periods is indicated in the right panels. Black lines indicate the distribution of isopycnals (σ$_\theta$) 25.6, 25.9, 26.2, 26.4, and 26.7 kg m$^{-3}$. Grey diamonds mark positions of CTD stations.**

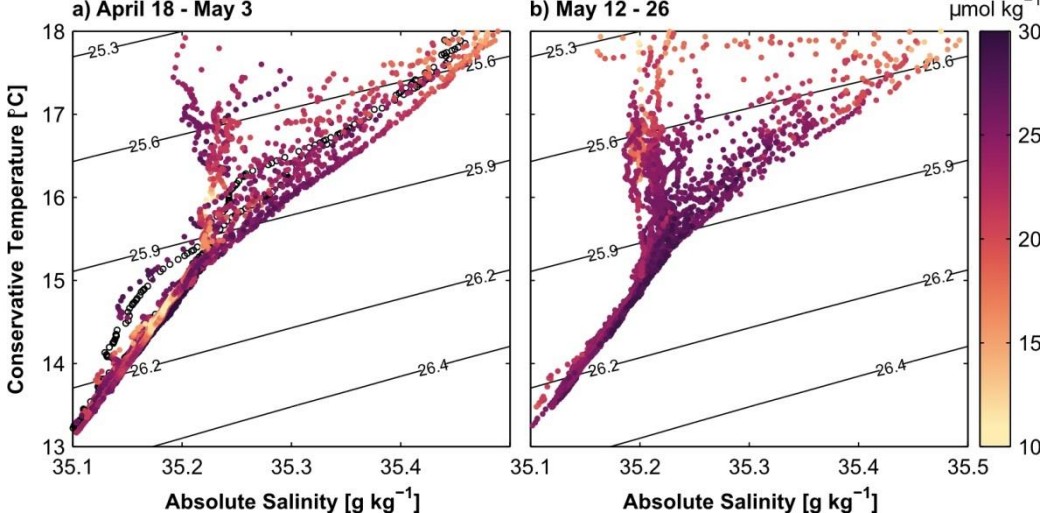

**Figure 10: Conservative temperature – absolute salinity diagram between 50 and 300 m depth for CTD profiles between the 100 and 400 m isobaths during April 18 – May 3, 2017 (left panel) and May 12 – 26, 2017 (right panel). Colour code depicts nitrate concentrations. Black contours indicate isopycnals ($\sigma_\theta$) in kg m$^{-3}$.**

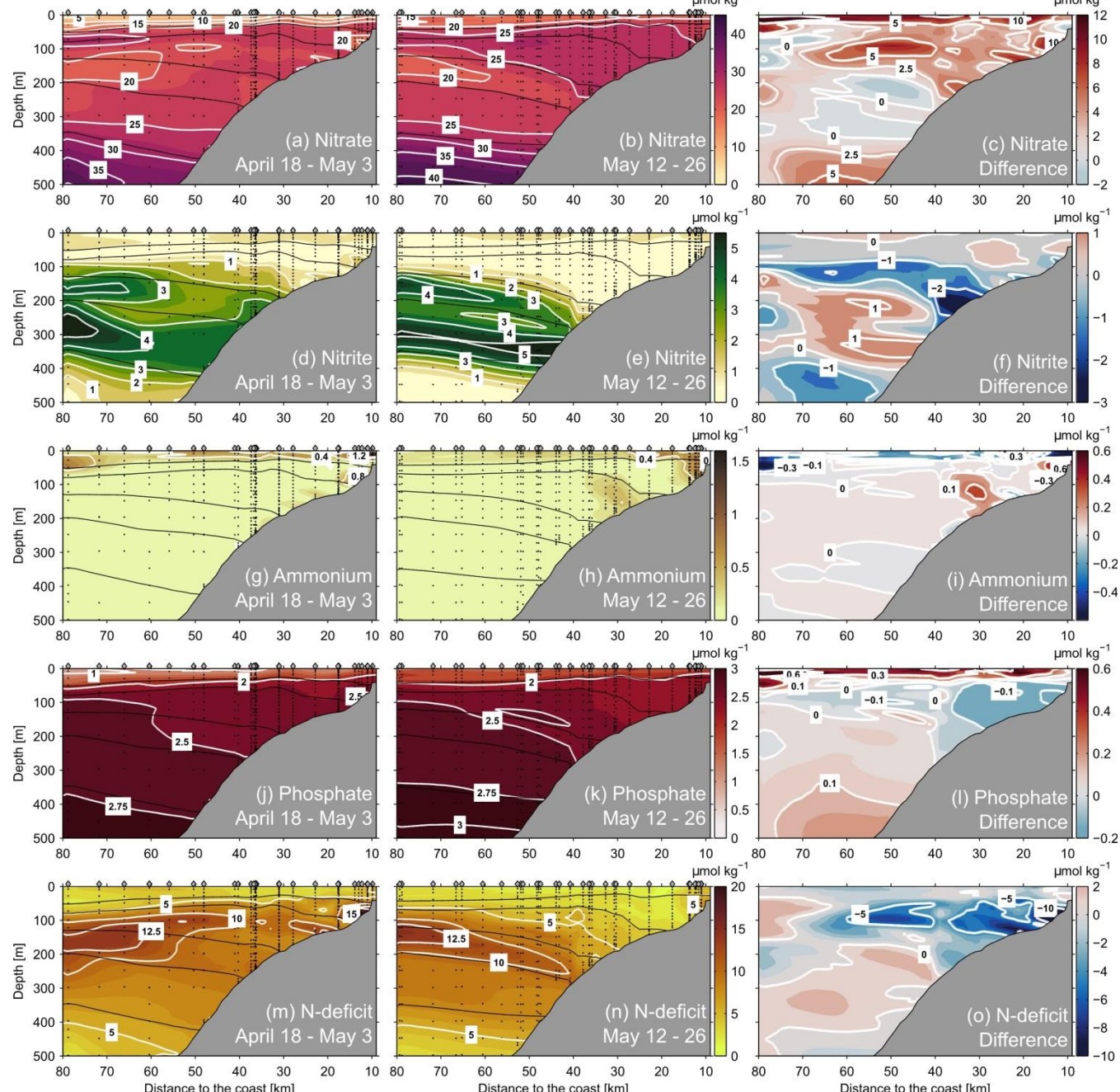

895

**Figure 11: Nitrate (upper panels), nitrite (2nd panel row), ammonium (3rd panel row), phosphate (4th panel row) concentrations, and nitrogen deficit (bottom panel row) at 12° S during April 18 – May 3, 2017 (left panels) and May 12 – May 26, 2017 (middle panels). Concentration difference of the respective parameters between the two time periods are shown in the left panels. Black lines indicate the distribution of isopycnals ($\sigma_\theta$) the 25.6, 25.9, 26.2, 26.4, and 26.7 kg m$^{-3}$. Grey diamonds mark the positions of CTD stations while**
900  **black circles mark the position of bottle samples in the water column.**