# Peer review of "Influence of intraseasonal eastern boundary circulation variability on hydrography and biogeochemistry off Peru"

_Ocean Science, 2019_

## Referee Comment (RC1) · Anonymous Referee #1 · 12 Sep 2019

General comments:

In this study, the authors analyse the alongshore coastal current intraseasonal variability and associated hydrodynamic and biogeochemical changes at 12S off the coast of Peru using in-situ measurements acquired during the mars-July 2017 shipboard sampling program. The manuscript shows some interesting results. It allows documenting the alongshore circulation during the propagation of a downwelling CTW mode 1 using real observations. The results are in agreement with recent modelling studies carried out over a longer period of time. Also, they look at the effect of the intensified poleward flow on the coastal hydrological and biogeochemical ocean properties. The

major weakness of the paper is that the authors can't extract the intraseasonal variations from the interannual/seasonal variations using the ship measurements making the interpretation of the results more difficult. I would advise the authors to provide a more throughout discussion of their results using remote sensed data (when possible). Also, having a model study will back up their hypothesis. Finally, one additional issue is that the manuscript is sloppily written (sometimes hard to get the sense), with clumsy wording and need proper editing. I will try to give some examples of this below. I will suggest to proof-read the manuscript by a native English speaker. Also, the figures are of good quality but the captions need to be reworked. Sentences in the captions are too long and some basics information are missing (full dates, units etc...). Overall, I find the manuscript worthy for publication in Ocean Science, after a major revision. Please find below a list of comments followed the flow of the text, that the authors should address in the revised manuscript.

Specific/technical comments:

1) In the Abstract, what is meant by less fixed nitrogen loss? I am not a biogeochemist and so I might misinterpret what is saying here but in reading this part of the paper (and also the conclusion), I got the impression that the change in Nitrate was driven by changes in the rate biogeochemical processes. But, later on (in reading the results and the discussion), this does not appear to be the case?

2) l21-22: The introduction could be better structured and clearer. As an example try to define Peru Upwelling System (PUS) at the beginning of your introduction.

3) l22: Fig1: Too many details. Simplify to highlight only the surface and subsurface current mentioned in the text.

4) l34, l52: Be careful when defining acronyms to put the initial letter of each word in Capital. Re-check all over the manuscript.

5) l48, l65, l168: timescales (try to be consistent over the manuscript)

6) l50-53: Explain more how local wind modulate the intraseasonal variability (upwelling intensity, local CTW..)

7) l52: Coastal Trapped Waves (CTW). Also, be consistent over the manuscript on the use of CTW or CTWs

8) l54: The sentence is incorrect, rephrase.

9) l55: Not reflected but transmitted. "Upon reaching the continental margin, part of the EKW bounces back along the equatorial waveguide into westward propagating equatorial Rossby waves, while part of incoming energy is transmitted poleward along the southwestern coast of South America as CTW".

10) l56: Poleward propagating CTW modulate the alongshore and vertical currents

11) l59: The author should mention the influence of the CTW on SLA as sea level data are used in the Results' section to track CTW

12) l60: this sentence is incorrect.

13) l62/63: The influence of remotely-forced CTW

14) l63: an individual propagating CTW

15) l67: Rephrase

16) In the Data section, the description of the ERSSTv5 data used in Figure 1 is missing.

17) l92- the investigation

18) l168- were smoothed

19) I will suggest to remove the position of the isopycnals from figure 2 and figure 3 as there are not useful for this section.

20) l209: Check the notation figure for Ocean Science: Fig. 2b and not Fig. 2(b)

21) l208 Sea surface height anomaly / SLA?

22) l211: The author use cm.s-1 in the text. Change Figure to put the unit in cm.s-1 to be consistent.

23) l218: At these depths and offshore

24) l225: data were

25) l242: the meaning is unclear

26) l232: The alongshore current modal structure of the three first modes

27) l227: Need to add the values of the velocity of the climatological alongshore current (Chaigneau et al., 2013) in the text which are necessary for the demonstration of the intensification of the PCUC. Also similarly to Figure 6 and figure 8, I would suggest adding a third panel to figure 3 to show the difference between the two periods of the amplitude of the alongshore current.

28) l233: And the phase speed? Are they consistent with the theoretical values obtained in Illig et al 2018a?

29) l236-237: Wouldn't it be more accurate to compare the CTW modal structures to the alongshore velocities anomaly (i.e the difference between the 2 periods chosen)?

30) l255: remove the second "and"

31) l257: How are the zonal equatorial and coastal alongshore wind anomalies? Here the authors could add the intraseasonal variations of the winds superimpose to Fig5 or in another figure.

32) l258: suggests

33) Section 4.3 and 4.4: This section describes the cross-shore structure of the hydrographic and biogeochemical tracers to the PCUC intensification during the two selected period. The text is too detailed which make it heavy to read. I think the major problem

with the writing overall is simply too many words. Please simply.

34) l264-265: You do not look at the processes in this study. Rephrase

35) l266-269: This is an example of sentences that could be shortened. Please rephrase

36) l277: I will found clearer to put the description related to the oxygen in section 4.4 (Response of the biogeochemical conditions to the PCUC intensification)

37) l285: Not many readers are familiar with the water masses characteristics off Peru. Please remind the reader what are the characteristics of the ESSW water masses (ESSW: T < 17 °C and S > 35; Silva and Neshyba, 1979; Chaigneau et al., 2013) in the text. Also, the authors may want to add a (small) panel with the position of the observations on a wider T/S diagram, with the T/S characteristics of the main water masses illustrated?

38) l267: dates on Figure 6 are wrong

39) l303: an increase of 5 ...

40) l358: The authors are looking at the total current vertical structure, not the anomaly.

41) l363: attributed to the second and third CTW modes. They found poleward velocities along the Peruvian.

42) l369: Could you explain how scattering will change the CTW vertical structure.

43) l371: the first CTW mode

44) l376: show

45) l378: alongshore winds

46) l383: the advection

47) l387-389: Total temperature/salinity cross-shore section during two periods (the

initial phase and the peak of the poleward flow) along with the differences between the two periods are shown in figures 6a-f. Figures show an increase in temperature and salinity along the continental shelf under 100m, which is in agreement with the stronger transport/advection of warmer and saltier ESSW, poleward. The surprise is that negative "anomalies (differences between the two periods)" in temperature (Fig. 6c) and salinity (Fig. 6f) are found in surface. I agree with the authors, this might result from the interannual/seasonal variations that have not been filtered from the signal analysed. However, this part of the discussion could be further elaborated (see general comment above), by (for example) looking at the SST from the satellites data from which intraseasonal variations can be estimated.

48) l411: Does N-loss means biogeochemical processes? Could the authors re-specified which ones and clarify this sentence?

49) l412: Why would you expect this? Is the increase of Nitrate (and then, the reduce nitrogen deficit) not related to the stronger transport poleward in the PUS of high nutrient ESSW as shown by Echevin et al., 2014? You may want to re-specified (or show?) the mean Nitrate characteristics and provide the mean alongshore gradient of Nitrate to support your demonstration.

50) l423-425: the meaning is unclear

51) l432: It won't be the case if equatorial waters were less rich in nutrients than the PUS. The sign of the anomaly depends on the sign of current anomaly and the sign of the gradient of the tracer (temperature or biogeochemical variables).

52) l436: due to

53) l438: Do the authors see changes in the coastal ecosystem? I wonder if the nutrient input associated with the downwelling CTW and the change in the N-P ratio is associated with a phytoplankton bloom as describe in Echevin et al., 2014. Have the authors looked at satellites chlorophyll data?

54) l442- suggests

55) l447- Again, at least, the small differences observed in temperature, salinity and oxygen could simply be due to the fact that the seasonal and interannual variations can not be removed from signal analysed. This statement is too strong for the conclusion.

56) l449/451: I do not think that was shown here (or at least not pointed out effectively). This study does not show changes in the rate of N-loss but rather point out the stronger transport of nitrate as the mechanisms for the nitrate/nitrogen deficit anomaly (in line with the results of Echevin et al., 2014). To look at how the biogeochemical cycles are affected by the CTW propagation further analysis are required (for example the use of a model). I will rephrase this to make your point clearer.

57) l452: "On intraseasonal timescales": From April to May 2017, our results suggest an increase in nitrate due to the passage of an intraseasonal downwelling CTW...

58) l453: A downwelling CTW?

59) l454 outcomes

60) l458: different from

---

## Referee Comment (RC2) · Anonymous Referee #2 · 2 Oct 2019

General comments: This research presents the shipboard currents, hydrographic and biogeochemical properties observation and satellite remote sensed sea surface temperature and sea level anomaly over the slope off the central Peru-Chile coasts. The authors used these data to investigate the possible scheme that determines shift of the upwelling system associated with the eastern boundary current in the southern hemisphere. Their most striking conclusion is that the southward propagating coastally-trapped waves (CTW), sourced from the equatorial current, played key (the authors used the word "likely" in the abstract) roles in determining those aforementioned variability in the upwelling system, or these CTW strengthened the southward transport of the sub-surface waters, which then "supersedes the simultaneous effect of down-

welling in terms of nutrient response". In my opinion, this conclusion is interesting, but still questionable, since the authors didn't provide sufficient solid analyses to support the schematic they drew in the abstract and conclusion section. Before presenting more specific comments, I have to admit that the results from their field measurements are invaluable and comprehensive, and the author put a lot of effort on the quality control and demonstrating them by using nice figures, although it took me some time to link the caption of those figures will the contents presented. Another great point of this research is that the authors did this research in a very interdisciplinary way. The combined discussion based on theories of physical and biogeochemical oceanography is very enlightening. The general comments, if I correctly summarized those specific ones, are that "the posted evidences cannot sufficiently support the conclusions" and "you need more evidences about the changes in the currents, not only in nutrient responses".

Specific Comments: 1) It is worthy for the authors to further polish their writing. The meaning of majority of those sentences is not easy to extract, since some sentences are too long and composed by many elements. I noticed that there is another published comment on the details about writing, and skipped them then. 2) The authors listed too many details in the data processing section without paying sufficient attention to the interlinkages among these data. Yes, processing data is important, but it is more important for the authors to guide us towards the mainstream of their research flow by introducing the procedure of data processing. I can just get what did you do, this or that, but cannot understand why did you do that. There are too many subsections in the section 2. Please also make sure that tides are not important in determining the general characteristics of the general circulation in your study area. 3) The introduction section is not well written either. The only points I can get are that the eastern boundary current and upwelling system experience multiscale variabilities that were not well studied, and the anomaly in winds (actually not only winds) can stimulate southward propagating CTW along the coastline. The authors didn't extract enough information from those cited historic studies to persuade us that CTW was found to greatly alter

the regional upwelling processes, for example, strong downwelling signal from historic studies was observed during upwelling-favorable forcing conditions. Those historic studies were just cited in and out without sufficient investigation. The novelty of this research is missing in this section, although it is much better summarized in the summary section. 4) I don't quite understand why did the authors link the effect of CTW to the intraseasonal variability of eastern boundary current, especially when they didn't do any analyses on the wind (stress and its curl) fields in the manuscript. Although they compared the observed currents with the climatological ones from, for example, numerical simulations, we still don't know whether the wind is comparable to is climatological conditions during the observation periods. Thereby, we cannot grantee that the variability is due to CTW, instead of migration of the wind system. Moreover, there were plenty of studies, for example, Zhang and Lentz [2017], have clearly showed that the response of shelf currents to the regional topography will also greatly modulate the domestic response of the current system. So, variability of the along-slope current itself is also worthy to be investigated. Talking about the time scale of intraseasonal, I also suggest the authors to investigate whether there are any meso-scale processes, for example, eddies, formed or detached from the main currents to generate the transition.

5) We knew that Kelvin waves or CTW will be continuously stimulated in its source region and propagate along the path you sketched. The authors used this process to explain the intraseasonal variability in the cold half year. Does that mean when the first CTW propagate through the system, the upwelling system will be shifted to a downwelling one and never switch back in the coming season? What will happen in, for example, December and January, when the downwelling system is switching back to an upwelling-dominant condition? It was also known that those CTW will be domestically arrested by irregularity of the along-slope topography to form standing waves and alter the regional cross-slope processes. The recent study of Kämpf [2018] also showed that there will be downstream propagation of topographic waves after the strong current passing through an irregular topography, for example, canyon or ridge. This is another possible process that determine the domestic response of the regional dynamics to the

[Figure]

CTW or general disturbances in both barotrophic and baroclinic modes.

In summary, this study is a great try to advance our understandings on the transition of the eastern boundary currents, and they provided us invaluable observational evidences and detailed analyses. However, it is not easy for this single research (not their series of studies) to answer all those previous questions. I suggest the authors to investigate the spatial and temporal variation of winds (stress and curl) and variability of the currents from, for example, numerical simulations or some widely used global simulations (e.g. HYCOM and CMEMS) to expand the vision of this research and make sure that the variability is mostly determined by the southward propagating CTW, instead of the other processes, including, for example, migration of wind system, along-shore variability of slope current and response of slope currents to the domestic irregular topography. The authors didn't show us the general distribution of the regional topography, yet. The authors are also suggested to more explicitly define the timescale of intraseasonal variability in the manuscript. In my opinion, CTW may determine the synaptic variation of the current system, while migration of the wind system (and the associate variation in the eastern boundary currents) will determine the entire background characteristics of the flow condition (upwelling or downwelling pattern). This will possibly be clearer than the term "intraseasonal" in your manuscript. A three-dimensional schematic of the flow pattern, propagation of CTW and responses in biogeochemical processes will greatly elevate this research, too.

References: Kämpf, J. (2018), On the Dynamics of Canyon–Flow Interactions, Journal of Marine Science and Engineering, 6(4), 129. Zhang, W., and S. J. Lentz (2017), Wind-driven circulation in a shelf valley. Part I: Mechanism of the asymmetrical response to along-shelf winds in opposite directions, Journal of Physical Oceanography, 47(12), 2927-2947.

---

## Author Comment (AC2) · 14 Feb 2020

**Reply to reviewer #2**

General comments: This research presents the shipboard currents, hydrographic and biogeochemical properties observation and satellite remote sensed sea surface temperature and sea level anomaly over the slope off the central Peru-Chile coasts. The authors used these data to investigate the possible scheme that determines shift of the upwelling system associated with the eastern boundary current in the southern hemisphere. Their most striking conclusion is that the southward propagating coastallytrapped waves (CTW), sourced from the equatorial current, played key (the authors used the word "likely" in the abstract) roles in determining those aforementioned variability in the upwelling system, or these CTW strengthened the southward transport of the subsurface waters, which then "supersedes the simultaneous effect of downwelling in terms of nutrient response". In my opinion, this conclusion is interesting, but still questionable, since the authors didn't provide sufficient solid analyses to support the schematic they drew in the abstract and conclusion section. Before presenting more specific comments, I have to admit that the results from their field measurements are invaluable and comprehensive, and the author put a lot of effort on the quality control and demonstrating them by using nice figures, although it took me some time to link the caption of those figures will the contents presented. Another great point of this research is that the authors did this research in a very interdisciplinary way. The combined discussion based on theories of physical and biogeochemical oceanography is very enlightening. The general comments, if I correctly summarized those specific ones, are that "the posted evidences cannot sufficiently support the conclusions" and "you need more evidences about the changes in the currents, not only in nutrient responses".

We would like to thank reviewer #2 for his/her encouraging but critical review of our manuscript and for the corrections and suggestions to improve the manuscript. We believe to have significantly improved the revised version of the manuscript upon his/her remarks and suggestions.

As detailed below, major changes in the revised version include a detailed analysis of the wind forcing and sea level anomaly using satellite observations, a rewritten introduction, an elaborate discussion of the possible impact of bathymetric features onto our alongshore flow observations, a more throughout discussion of the results and a better reasoning as far biogeochemical processes are concerned. Finally, we polished the writing of the text and the figure captions.

In our detailed response below, comments by the reviewer are in bold letters and changes in the manuscript are expressed in italic letter. We belief that some of the reviewer comments originated from sections where our writing was misleading. We hope to have corrected that in the revised version.

**Specific Comments:**

1) It is worthy for the authors to further polish their writing. The meaning of majority of those sentences is not easy to extract, since some sentences are too long and composed by many elements. I noticed that there is another published comment on the details about writing, and skipped them then.

We made great effort to improve the writing and comprehensibility of the manuscript. Please accept our apologies for not having done that before submitting the first version.

2) The authors listed too many details in the data processing section without paying sufficient attention to the interlinkages among these data. Yes, processing data is important, but it is more important for the authors to guide us towards the mainstream of their research flow by introducing the procedure of data processing. I can just get what did you do, this or that, but cannot understand why did you do that. There are too many subsections in the section 2. Please also make sure that tides are not important in determining the general characteristics of the general circulation in your study area.

Thank you for this comment. We significantly shortened the data and methods section wherever possible and tried to motivate our use of processing and analysis techniques. However, we think that our brief descriptions in the data and method sections are necessary to allow replicability of our results from the published data. The different subsections in section 2 and 3 will allow the reader to extract specific information on data or methods without needing to go through longer data or methods section. We thus decided to retain most subsections in section 2 and 3.

3) The introduction section is not well written either. The only points I can get are that the eastern boundary current and upwelling system experience multiscale variabilities that were not well studied, and the anomaly in winds (actually not only winds) can stimulate southward propagating CTW along the coastline. The authors didn't extract enough information from those cited historic studies to persuade us that CTW was found to greatly alter the regional upwelling processes, for example, strong downwelling signal from historic studies was observed during upwelling-favorable forcing conditions. Those historic studies were just cited in and out without sufficient investigation. The novelty of this research is missing in this section, although it is much better summarized in the summary section.

We agree with your comment. In the revised version, the introduction was reorganized and rewritten. We now focus on describing the Peruvian upwelling system, the consequences of variable nutrient and oxygen availability on biogeochemical processes, local and remote forcing of intraseasonal flow variability including effects of variable topography and the impact of intraseasonal flow variability on biogeochemistry. While doing so, we build upon historical studies. Finally, we improved the motivation of our study.

4) I don't quite understand why did the authors link the effect of CTW to the intraseasonal variability of eastern boundary current, especially when they didn't do any analyses on the wind (stress and its curl) fields in the manuscript. Although they compared the observed currents with the climatological ones from, for example, numerical simulations, we still don't know whether the wind is comparable to is climatological conditions during

**the observation periods. Thereby, we cannot grantee that the variability is due to CTW, instead of migration of the wind system.**

Thank you for pointing this out. We addressed this comment by including a detailed analysis of the wind variability prior and during the observed strengthening of the poleward boundary current flow. Additionally, we added a discussion of possible local wind forcing mechanisms in the introduction. Finally, we included an analysis of equatorial winds that triggered an equatorial Kelvin wave. In the results section of the manuscript, we added two subsections analyzing local and remote winds:

**4.2.1 Role of local wind stress**

A potential local forcing mechanism of the intensified PCUC flow are anomalies of local wind stress curl. An increase in the magnitude of near-coastal negative wind stress curl leads to increased poleward flow along the eastern boundary through Sverdrup dynamics (e.g., Marchesiello et al., 2003). The adjustment of the circulation to changes in the wind stress curl at the eastern boundary is rather fast and occurs within a few days (Klenz et al., 2018). Wind stress curl along the Peruvian continental margin between 10° S and 14° S was negative throughout the observational period (Fig. 4), continuously forcing poleward flow. However, during the period of PCUC acceleration between end of April and mid-May, the magnitude of negative wind stress curl decreased (Fig. 4c, d, e, f). It can thus be ruled out that local wind stress curl forcing is responsible for the observed intensified PCUC. Nevertheless, elevated negative wind stress curl was observed from May 18 – 22, which may have contributed to maintaining a strong PCUC in late-May.

Variability of near-coastal alongshore wind stress excites CTWs which propagate poleward (e.g. Yoon and Philander, 1982) and thereby enhance or decrease poleward flow within the depth range of the PCUC. Model studies show that CTWs are excited near the equatorward edge of the region of wind variability (e.g. Fennel et al., 2012). In Mid-April through May 2017, alongshore wind stress between 6°S and 15°S was variable (Fig. 5). While moderate wind stress (0.03-0.06 N m-2) prevailed from mid-April to May 3, it was weak during the first two weeks of May (Fig. 5d,e, g). However, during the later period the strong acceleration of the poleward flow occurred, requiring an intensification of alongshore wind stress. Thus, the initial acceleration of the PCUC during this period (Fig. 2d, e) cannot be related to local wind stress variability. Alongshore wind stress did significantly strengthen on May 15 and remained elevated for a period of about 5 days. This wind event was intense between 15° and 8° S, but did not occur north of 8° S. CTWs were likely excited in the region between 12° and 8° S that contributed to the elevated poleward velocities observed in the later phase between May 17 and 26 (Fig. 2f). 4.2.2 Equatorial winds and wave response.

A weakening of the trade winds at the equator by e.g. westerly wind events forces downwelling on the equator which in turn generates an eastward propagating equatorial Kelvin waves, which in turn may have transmitted parts of its energy to a CTW at the eastern boundary. Indeed, several westerly wind anomalies occurred in the central and eastern equatorial Pacific during the first 6 month of 2017 (Fig. 6). A particularly elevated westerly wind anomaly between the date line and 120° W occurred during the first two weeks of April (Fig. 7a). A positive SLA propagating along the equator appears to the east of the wind event at about 100° W (Fig. 7b). Moreover, there were plenty of studies, for example, Zhang and Lentz [2017], have clearly showed that the response of shelf currents to the regional topography will also greatly modulate the domestic response of the current system. So, variability of the along-slope current itself is also worthy to be investigated.

We fully agree. However, as written above, the changes in local winds did not force the accelerated Peru-Chile Undercurrent. We added a discussion on the effect of variable bathymetry to section 6 (please also see our response to 5) below).

**Talking about the time scale of intraseasonal, I also suggest the authors to investigate whether there are any meso-scale processes, for example, eddies, formed or detached from the main currents to generate the transition.**

Our data set collected during the cruise as well as SLA data from satellites did not indicate any mesoscale eddy generation during the alongshore flow acceleration period. This argument also hold for the period of elevated flow from May 17 to May 26.

5) We knew that Kelvin waves or CTW will be continuously stimulated in its source region and propagate along the path you sketched. The authors used this process to explain the intraseasonal variability in the cold half year. Does that mean when the first CTW propagate through the system, the upwelling system will be shifted to a downwelling one and never switch back in the coming season? What will happen in, for example, December and January, when the downwelling system is switching back to an upwellingdominant condition?

Thank you for pointing out difficulties in understanding our previous version of the manuscript. It was not our intention to argue that the described CTW is changing the state of the upwelling system. Instead, we use the term "downwelling CTW" exclusively to define the sign of velocity and SLA anomaly. In our definition, a downwelling CTW depresses the thermocline and is associated with an increase of SLA near the coast and enhanced poleward flow. However, as we also state in the manuscript, near-coastal surface temperatures decrease during the passing of the downwelling CTW. In more general terms, we cannot conclusively determine the impact of the CTW on the upwelling system itself. The decreased SSTs near the coast were not associated with enhanced but with declining chlorophyll concentrations. It is thus likely that elevated local wind from May 15 - 20 enhanced near-surface heat loss leading to cooling of the top few meters of the coastal water column. We added parts of this discussion to section 4.1 of the manuscript, where the near-surface cooling during the CTW event is mentioned.

The focus of our manuscript is on the variability of hydrography, oxygen and nutrient distributions in the upper thermocline of the Peruvian upwelling system. This depth range often lacks oxygen and variability of nutrients and oxygen here is very relevant for biogeochemical processes. We hope to have improved the focus of the paper by restructuring the introduction and by improving the discussion in the manuscript.

It was also known that those CTW will be domestically arrested by irregularity of the along-slope topography to form standing waves and alter the regional cross-slope processes. The recent study of Kämpf [2018] also showed that there will be downstream propagation of topographic waves after the strong current passing through an irregular

**topography, for example, canyon or ridge. This is another possible process that determine the domestic response of the regional dynamics to the CTW or general disturbances in both barotrophic and baroclinic modes.**

We thank the reviewer for pointing out that a discussion of the impact of irregular along-slope topography on the observed flow variability was missing in the previous version of the manuscript. In the revised version, we enlarged the insert in Fig.1 showing the distribution of topography between 10° S and 15° S and discuss topographic features near our sampling site. There is a small ridge to the north and the shelf narrows south of our sampling site. However, other than that there are no elevated topographic irregularities such as canyons. Nevertheless, CTW scattering at the upstream ridge can potentially increase the flow at our sampling site (Wang, 1980; Wilkin and Chapman, 1990). The narrowing of the shelf further downstream may also potentially influence the upstream circulation (e.g. as described by Wilkin and Chapman, 1990). However, we also point out that observations and models suggest that equatorially-forced first mode CTWs along the South American coast propagate past 25° S (e.g. Shaffer et al., 1997, Illig et al., 2018). As discussed by Illig et al (2018) in terms of the Burger number variability (Huthnance, 1978) the effect of stratification on the CTW parameters is found to be more important than irregular along-slope topography in the region of our sampling location. We added the following paragraph to the discussion in our manuscript:

Local bathymetry interacts with the passing CTW as well. North of our sampling site the continental slope bends offshore at depths between 500 m to 1000 m (Fig. 1, insert) and the shelf narrows south. Changes in coastline, shelf width, and along-slope bathymetry leads to a transfer of CTW energy into higher modes (scattering) and upstream backscattering (Wang, 1980; Wilkin and Chapman, 1990; Kämpf (2018); Brunner et al., 2019). The influence of the changes in shelf width on the upstream alongshore flow structure can extend to 200 km upstream (Wilkin and Chapman, 1990). Furthermore, the bent of the continental slope north of out sampling site may lead to CTW scattering which may additionally intensify the poleward flow at our sampling site. A recent model study suggests that differences between the theoretical CTW solutions and observations are predominately due to wave scattering (Brunner et al., 2019).

In summary, this study is a great try to advance our understandings on the transition of the eastern boundary currents, and they provided us invaluable observational evidences and detailed analyses. However, it is not easy for this single research (not their series of studies) to answer all those previous questions. I suggest the authors to investigate the spatial and temporal variation of winds (stress and curl) and variability of the currents from, for example, numerical simulations or some widely used global simulations (e.g. HYCOM and CMEMS) to expand the vision of this research and make sure that the variability is mostly determined by the southward propagating CTW, instead of the other processes, including, for example, migration of wind system, along-shore variability of slope current and response of slope currents to the domestic irregular topography. The authors didn't show us the general distribution of the regional topography, yet. The authors are also suggested to more explicitly define the timescale of intraseasonal variability in the manuscript. In my opinion, CTW may determine the synaptic variation in the

eastern boundary currents) will determine the entire background characteristics of the flow condition (upwelling or downwelling pattern). This will possibly be clearer than the term "intraseasonal" in your manuscript. A three-dimensional schematic of the flow pattern, propagation of CTW and responses in biogeochemical processes will greatly elevate this research, too.

We thank the reviewer for providing very valuable comments and suggestions for improving our manuscript. As detailed above, we considered most of his/her corrections and suggestions. We think that by including additional analysis of winds, sea level anomaly and irregular topography in the revised version of the manuscript, we provide sufficient evidence for understanding the nature of the observed flow intensification. Thus, we refrained from looking into global simulations such as HYCOM or CMEMS.

References: Kämpf, J. (2018), On the Dynamics of Canyon–Flow Interactions, Journal of Marine Science and Engineering, 6(4), 129.

**Zhang, W., and S. J. Lentz (2017), Wind-driven circulation in a shelf valley. Part I: Mechanism of the asymmetrical response to along-shelf winds in opposite directions, Journal of Physical Oceanography, 47(12), 2927-2947.**

Huthnance, J. M.: On Coastal Trapped Waves: Analysis and Numerical Calculation by Inverse Iteration, J. Phys. Oceanogr., 8(1), 74–92, doi:10.1175/1520-0485(1978)0082.0.CO;2, 1978.

Illig, S., Bachèlery, M.-L. and Cadier, E.: Subseasonal Coastal-Trapped Wave Propagations in the Southeastern Pacific and Atlantic Oceans: 2. Wave Characteristics and Connection With the Equatorial Variability, J. Geophys. Res. Oceans, 123(6), 3942–3961, doi:10.1029/2017JC013540, 2018.

Wang, D.-P.: Diffraction of Continental Shelf Waves by Irregular Alongshore Geometry, J. Phys. Oceanogr., 10(8), 1187–1199, doi:10.1175/1520-0485(1980)0102.0.CO;2, 1980.

Wilkin, J. L. and Chapman, D. C.: Scattering of Coastal-Trapped Waves by Irregularities in Coastline and Topography, J. Phys. Oceanogr., 20(3), 396–421, doi:10.1175/1520-0485(1990)0202.0.CO;2, 1990.

---

## Author Comment (AC1)

**Reply to reviewer #1:**

**General comments:**

In this study, the authors analyse the alongshore coastal current intraseasonal variability and associated hydrodynamic and biogeochemical changes at 12S off the coast of Peru using in-situ measurements acquired during the mars-July 2017 shipboard sampling program. The manuscript shows some interesting results. It allows documenting the alongshore circulation during the propagation of a downwelling CTW mode 1 using real observations. The results are in agreement with recent modelling studies carried out over a longer period of time. Also, they look at the effect of the intensified poleward flow on the coastal hydrological and biogeochemical ocean properties. The major weakness of the paper is that the authors can't extract the intraseasonal variations from the interannual/seasonal variations using the ship measurements making the interpretation of the results more difficult. I would advise the authors to provide a more throughout discussion of their results using remote sensed data (when possible). Also, having a model study will back up their hypothesis. Finally, one additional issue is that the manuscript is sloppily written (sometimes hard to get the sense), with clumsy wording and need proper editing. I will try to give some examples of this below. I will suggest to proofread the manuscript by a native English speaker. Also, the figures are of good guality but the captions need to be reworked. Sentences in the captions are too long and some basics information are missing (full dates, units etc...). Overall, I find the manuscript worthy for publication in Ocean Science, after a major revision. Please find below a list of comments followed the flow of the text, that the authors should address in the revised manuscript.

We would like to thank reviewer #1 for his/her critical review of our manuscript as well as the corrections and suggestions to improve the manuscript. We believe to have significantly improved the revised version of the manuscript upon his/her remarks and suggestions.

As detailed below, major changes in the revised version include a detailed analysis of the wind forcing and sea level anomaly using satellite observations, a rewritten introduction, a more throughout discussion of the results and a better reasoning as far biogeochemical processes are concerned. We also corrected sloppy wording and figure captions.

In our detailed response below, comments by the reviewer are in bold letters and changes in the manuscript are expressed in italic letter.

**Specific/technical comments:**

1) In the Abstract, what is meant by less fixed nitrogen loss? I am not a biogeochemist and so I might misinterpret what is saying here but in reading this part

**of the paper (and also the conclusion), I got the impression that the change in Nitrate was driven by changes in the rate biogeochemical processes. But, later on (in reading the results and the discussion), this does not appear to be the case?**

Thank you for pointing this out. Indeed, the increase in nitrate is not due to changes in the rate of biogeochemical processes but due to the shortened residence time of the water masses in the Peruvian Upwelling System (PUS). The shortened residence time in turn is due to increased alongshore advection of the water masses within the Peru-Chile Undercurrent. We have reformulated the abstract and explain this in more detail later in the manuscript to be more clear about this result. The abstract now reads

An intensified poleward flow increases water mass advection from the equatorial current system to the study site. The impact of the elevated advection was mostly noticed in the nitrogen cycle. Shorter transit times between the equator and the coast off central Peru led to a strong increase in nitrate concentrations, less fixed nitrogen loss to  $N_2$ , and a decrease in the nitrogen deficit.

**2) I21-22: The introduction could be better structured and clearer. As an example try to define Peru Upwelling System (PUS) at the beginning of your introduction.**

Thank you. We now define the Peru Upwelling System at the beginning of the introduction. Additionally, we restructured the introduction at several places to improve its clarity. The introduction now starts as

The Peruvian Upwelling System (PUS) is one of the biologically most productive regions in the world's ocean resulting in economically important fish catches (e.g. Carr, 2002; Chavez et al., 2008). Located in the Eastern Tropical South Pacific (ETSP), the high surface productivity of the PUS is most pronounced within 100 km off the Peruvian coast between 4 and 16° S (Pennington et al., 2006).

**3) I22: Fig1: Too many details. Simplify to highlight only the surface and subsurface current mentioned in the text.**

Agreed, we removed from figure 1 currents not mentioned in the text.

**4) I34, I52: Be careful when defining acronyms to put the initial letter of each word in Capital. Re-check all over the manuscript.**

Thank you, we have changed the spelling of words when introducing acronyms according to the suggestions.

Located in the Eastern Tropical South Pacific (ETSP), the high surface productivity ... (Note that this acronym is now defined in the first paragraph due to the restructuring of the introduction.)

**5) I48, I65, I168: timescales (try to be consistent over the manuscript)**

Thank you for pointing out the inconsistent spelling. We changed *time scales* to *timescales* throughout the manuscript.

**6) I50-53: Explain more how local wind modulate the intraseasonal variability (upwelling intensity, local CTW..)**

Thank you, we agree with your comment and added a paragraph to the introduction explaining how local wind variability leads to variability of the PCUC. We also added a detailed analysis of local and remote wind forcing to the manuscript as suggested in your later comments. In the introduction we now state:

Intraseasonal variability of the eastern boundary circulation is either forced locally by changes of the wind system above the PUS or forced remotely by variability of the wind system at the equator. Strengthening (weakening) of the local alongshore winds causes intensified (reduced) Ekman divergence close to the coast that accelerates (decelerates) the coastal surface jet (e.g. Philander and Yonn, 1982; Yoon and Philander, 1982; McCreary et al., 1987; Fennel et al., 2012). At the same time, Coastal Trapped Waves (CTWs) are excited propagating poleward to set up an alongshore pressure gradient balancing the accelerating (decelerating) alongshore flow (Yoon and Philander, 1982). Due to differences in vertical structure of the surface jet and the excited CTWs, the poleward flowing PCUC accelerates (decelerates). Likewise, variability of local wind stress curl forces variability of the poleward undercurrent through enhancing or reducing Sverdrup transport in the eastern boundary current regime (e.g. McCreary and Chao, 1985; Marchesiello et al., 2003; Junker et al., 2015; Klenz et al., 2018). Local wind-forced variability of eastern boundary poleward undercurrents has been reported from the Californian, Mauritanian and Benguela eastern boundary upwelling regions on time scales from intraseasonal to seasonal (e.g. Allen and Smith, 1981; Marchesiello et al., 2003; Junker et al., 2015; Klenz et al., 2018).

**7) I52: Coastal Trapped Waves (CTW). Also, be consistent over the manuscript on the use of CTW or CTWs**

Thank you. We are now consistent in using CTW/CTWs over the manuscript.

**8) I54: The sentence is incorrect, rephrase.**

Indeed. We have rephrased the sentence ("Wind events in the equatorial Pacific can generate equatorial Kelvin waves that propagate eastward ") to

Variability of zonal winds in the equatorial Pacific forces equatorial Kelvin waves that propagate eastward.

9) I55: Not reflected but transmitted. "Upon reaching the continental margin, part of the EKW bounces back along the equatorial waveguide into westward propagating equatorial Rossby waves, while part of incoming energy is transmitted poleward along the southwestern coast of South America as CTW".

Yes, thank you. We have changed the sentence to

Upon reaching the continental margin, part of their incoming energy is transmitted poleward along the southwestern coast of South America as CTWs.

We did not include the Rossby wave part, as their frequency range is limited and particular the energy of high frequency Kelvin waves (i.e. with periods from a week to a month) cannot bounce back into westward propagating equatorial Rossby waves.

**10) I56: Poleward propagating CTW modulate the alongshore and vertical currents**

Thank you, we extensively revised this part of the introduction. The first sentence of this paragraph now states

While modulating the alongshore circulation and vertical velocities, poleward propagating CTWs produce vertical displacements of the pycnocline of the order of tens of meters and sea level changes of a few centimeters (Leth and Middelton, 2006, Colas et al., 2008, Belmadani et al., 2012).

**11) I59: The author should mention the influence of the CTW on SLA as sea level data are used in the Results' section to track CTW**

Indeed, please see our reply to comment 10) above.

**12) I60: this sentence is incorrect.**

We agree. We altered this statement to be clear that local wind forcing and heat fluxes are the most important factors for intraseasonal SST variability and CTW contribution is of less importance.

On the other hand, intraseasonal sea surface temperature (SST) variability is suggested to be mainly driven by local winds and heat fluxes while CTWs play only a minor role (Dewitte et al., 2011; Illig et al., 2014).

**13) I62/63: The influence of remotely-forced CTW**

Changed

**14) I63: an individual propagating CTW**

Changed

**15) I67: Rephrase**

This part of the introduction was completely rewritten and this sentence is no longer in the text.

**16) In the Data section, the description of the ERSSTv5 data used in Figure 1 is missing.**

We have added a paragraph on the ERSST data to section 2.

Sea surface temperature from the NOAA Extended Reconstructed Sea Surface Temperature, Version 5 (ERSSTv5) dataset (Huang et al., 2017a) was used. This dataset provides monthly

values of SST on a 2°x2° grid based on in-situ temperature observations from several sources (Huang et al., 2017b).

**17) 192- the investigation**

Changed.

**18) I168- were smoothed**

Changed.

**19) I will suggest to remove the position of the isopycnals from figure 2 and figure 3 as there are not useful for this section.**

Indeed, the isopycnals do not contribute to the analysis of circulation variability in section 4.1. However, we find it helpful for comparing the velocity distributions with the hydrographic sections and nutrient distributions shown in figures 8 and 10. These figures depict the same isopycnal surfaces. Therefore, we decided to keep the isopycnals in the figures.

**20) I209: Check the notation figure for Ocean Science: Fig. 2b and not Fig. 2(b)**

Thank you for this comment. We have changed the notation throughout the manuscript.

**21) I208 Sea surface height anomaly / SLA?**

Indeed, we have changed *sea surface height anomaly* to *Sea Level Anomaly (SLA)* and consistently use the abbreviation throughout the manuscript.

**22) I211: The author use cm.s-1 in the text. Change Figure to put the unit in cm.s-1 to be consistent.**

Thank you for pointing this out. We have changed the units of velocity to m s-1 throughout the manuscript and are now consistent with the units presented in the figures.

**23) I218: At these depths and offshore**

Changed.

**24) I225: data were**

Changed as well.

**25) I242: the meaning is unclear**

Thank you. This section was greatly revised. We now discuss alongshore velocity differences between two periods instead of alongshore velocity itself (also see comment 29) below. The sentence in question referring to a possible superposition of equatorward flow due to the Chile-Peru Deep Coastal Current and the CTW velocity signal is thus obsolete and was removed from the manuscript.

26) I232: The alongshore current modal structure of the three first modes

**Changed**

27) I227: Need to add the values of the velocity of the climatological alongshore current (Chaigneau et al., 2013) in the text which are necessary for the demonstration of the intensification of the PCUC. Also similarly to Figure 6 and figure 8, I would suggest adding a third panel to figure 3 to show the difference between the two periods of the amplitude of the alongshore current.

Thank you for this suggestion. We added a third panel showing the difference in velocity between both time periods to figure 3. Also, we added a sentence on the climatological alongshore velocities from Chaigneau et al. (2013) to facilitate an evaluation of the observed flow intensification.

The intensified PCUC flow strongly exceeds climatological PCUC flow reported from this region. Mean alongshore flow at 12° S determined from vmADCP data sampled during 22 cruises show maximum PCUC core velocities of  $0.1 - 0.15 \text{ m s}^{-1}$  (Chaigneau et al., 2013), similar to the situation observed during April 18-25 and June 24 (Fig. 2b and g).

**28) I233: And the phase speed? Are they consistent with the theoretical values obtained in Illig et al 2018a?**

The phase speeds of the first two modes are consistent with the values obtained by Illig et al. (2018a). However, the phase speed of the third mode is slightly below the range of their reported values. In the text, we now state:

The obtained phase speeds are within the ranges reported by Illig et al. (2018a, their table 1) for the first two modes while the phase speed of the third mode is slightly lower than their results (0.82 ms-1 compared to  $0.93\pm0.08$  ms-1). The velocity structure of the modes is very similar to their structure reported in their study at 16°S as well.

The structure of section 4. Results was greatly revised. The CTW modes and their phase speeds are now discussed in section 4.1.3 Modal structure of the intensified flow.

**29) I236-237: Wouldn't it be more accurate to compare the CTW modal structures to the alongshore velocities anomaly (i.e the difference between the 2 periods chosen)?**

Indeed, thank you for this comment. In the revised version, we compare the modal structures to the difference of alongshore velocity between the two time periods and interpret it accordingly. We have modified Fig. 5 accordingly and adapted the text in section 4.1.3 and in the discussion.

**30) I255: remove the second "and"**

Removed.

**31) I257: How are the zonal equatorial and coastal alongshore wind anomalies? Here the authors could add the intraseasonal variations of the winds superimpose to Fig5 or in another figure.**

We thank the reviewer for this comment. In the revised version, we are including a detailed analysis of local and equatorial wind stress and wind stress curl variability in the results section and include a discussion of the results in the section 5. Summary and discussion. We show that local wind stress and wind stress curl forcing cannot explain the intensification of the PCUC. However, CTWs excited by local wind stress and Sverdrup transport likely have contributed to enhancing poleward flow later in May. We now show that an elevated westerly wind burst at the equator during April 2017 is likely responsible for the generation of a Kelvin wave that reached the eastern boundary beginning of May 2017. Here, it transmitted parts of its energy to a CTW moving poleward along the continental margin of South America. We have added the following sections to the results:

**4.2.1 Role of local wind stress**

A potential local forcing mechanism of the intensified PCUC flow are anomalies of local wind stress curl. An increase in the magnitude of near-coastal negative wind stress curl leads to increased poleward flow along the eastern boundary through Sverdrup dynamics (e.g., Marchesiello et al., 2003). The adjustment of the circulation to changes in the wind stress curl at the eastern boundary is rather fast and occurs within a few days (Klenz et al., 2018). Wind stress curl along the Peruvian continental margin between  $10^{\circ}$  S and  $14^{\circ}$  S was negative throughout the observational period (Fig. 4), continuously forcing poleward flow. However, during the period of PCUC acceleration between end of April and mid-May, the magnitude of negative wind stress curl decreased (Fig. 4c, d, e, f). It can thus be ruled out that local wind stress curl forcing is responsible for the observed intensified PCUC. Nevertheless, elevated negative wind stress curl was observed from May 18 - 22, which may have contributed to maintaining a strong PCUC in late-May.

Variability of near-coastal alongshore wind stress excites CTWs which propagate poleward (e.g. Yoon and Philander, 1982) and thereby enhance or decrease poleward flow within the depth range of the PCUC. Model studies show that CTWs are excited near the equatorward edge of the region of wind variability (e.g. Fennel et al., 2012). In Mid-April through May 2017, alongshore wind stress between 5° S and 15° S was variable (Fig. 5). While moderate wind stress (0.03-0.06 N m-2) prevailed from mid-April to May 3, it was weak during the first two weeks of May (Fig. 5d, e, g). However, during the later period the strong acceleration of the poleward flow occurred, requiring an intensification of alongshore wind stress. Thus, the initial acceleration of the PCUC during this period (Fig. 2d, e) cannot be related to local wind stress variability. Alongshore wind stress did significantly strengthen on May 15 and remained elevated for a period of about 5 days. This wind event was intense between 15° and 5° S, but did not occur north of 8° S. CTWs were likely excited in the region between 12° and 8° S that contributed to the elevated poleward velocities observed in the later phase between May 17 and 26 (Fig. 2f).

**4.2.2 Equatorial winds and wave response.**

A weakening of the trade winds at the equator by e.g. westerly wind events forces downwelling on the equator which in turn generates an eastward propagating equatorial Kelvin waves, which in turn may have transmitted parts of its energy to a CTW at the eastern boundary. Indeed, several westerly wind anomalies occurred in the central and eastern equatorial Pacific during the first 6 month of 2017 (Fig. 6). A particularly elevated westerly wind anomaly between the date line and 120° W occurred during the first two weeks of April (Fig. 6a). At the same time, a positive SLA propagating along the equator appears to the east of the wind event at about 100° W (Fig. 6b).

**32) I258: suggests**

Changed.

33) Section 4.3 and 4.4: This section describes the cross-shore structure of the hydrographic and biogeochemical tracers to the PCUC intensification during the two selected period. The text is too detailed which make it heavy to read. I think the major problem with the writing overall is simply too many words. Please simply.

We have shortened and simplified the manuscript throughout these sections.

**34) I264-265: You do not look at the processes in this study. Rephrase**

Thank you for pointing this out. We have rephrased this sentence to

In the following we analyse the changes in hydrographic conditions co-occurring with the increase of alongshore flow.

**35) I266-269: This is an example of sentences that could be shortened. Please rephrase**

We significantly shortened this section and removed repeated and unnecessary information. Former lines 266-269 now read:

Lower near-surface conservative temperatures near the coast compared to offshore (Fig. 8a, b) indicated active upwelling during the observational program. While the upwelling signal was restricted to the upper 50 m, near-coastal water masses between 50 and 300m were significantly warmer compared to water masses offshore (8a).

**36) I277: I will found clearer to put the description related to the oxygen in section 4.4 (Response of the biogeochemical conditions to the PCUC intensification)**

We agree that oxygen concentration may also be classified as a biogeochemical parameter. However, in the manuscript we discuss oxygen distributions analogously to temperature and salinity while in section 4.4 (discussion of biogeochemical parameters) we describe nutrient distributions and related parameters. The later discussion does not require linking to the distribution of oxygen concentrations. For us, it thus seems more clear to keep the current subsection separation. However, to make clearer that section 4.4 is restricted to the nutrient biogeochemistry we rename it as *"4.4 Response of nutrient biogeochemistry to the PCUC intensification"*.

37) I285: Not many readers are familiar with the water masses characteristics off Peru. Please remind the reader what are the characteristics of the ESSW water masses (ESSW: T < 17 $\circ$ C and S > 35; Silva and Neshyba, 1979; Chaigneau et al., 2013) in the text. Also, the authors may want to add a (small) panel with the position of the observations on a wider T/S diagram, with the T/S characteristics of the main water masses illustrated?

Thank you, we have added the T and S characteristics of ESSW and ESPIW to the text, in both cases we have relied on the most recent classification by Grados et al. (2018).

ESSW originates from the equatorial current system. It is characterized by a linear relationship of temperature and salinity in the temperature range 8 - 14 °C and absolute salinity range 34.6 - 35.0 (e.g. Grados et al., 2018). Lower salinity Eastern South Pacific Intermediate Water (temperature range 12 - 14 °C, salinity 34.8 (Grados et al., 2018)), which is also situated in the depth range mentioned above was only observed in the hydrographic data from two offshore stations (Fig. 10a).

**38) I267: dates on Figure 6 are wrong**

Thank you. An erroneous date was actually inserted in former Figure 3 and 7. We corrected the dates in the figures throughout the manuscript.

**39) 1303: an increase of 5 ...**

Changed.

**40) I358: The authors are looking at the total current vertical structure, not the anomaly.**

Indeed, thank you. In the revised version we now discuss velocity anomalies that are presented in Figure 3 as suggested by the reviewer. We now highlight our hypothesis that the increase of PCUC velocities in May relative to April was caused by a CTW.

... that occurred simultaneously to the increase of poleward flow at 12 °S.

**41) I363: attributed to the second and third CTW modes. They found poleward velocities along the Peruvian.**

Changed accordingly.

**42) I369: Could you explain how scattering will change the CTW vertical structure.**

Scattering of the wave at changing topography will transmit the energy into higher modes. We included a brief explanation in the discussion:

The local topography interacts with the passing CTW as well and such an interaction is neglected in the CTW mode solutions derived here. [...] At changes in the topography parts of the wave energy can be transmitted into higher modes or upstream scattering (Wang, 1980; Wilkin and Chapman, 1990; Brunner et al., 2019).

**43) I371: the first CTW mode**

Changed.

44) 1376: show

Changed.

45) I378: alongshore winds

Changed.

46) 1383: the advection

Changed.

47) I387-389: Total temperature/salinity cross-shore section during two periods (the initial phase and the peak of the poleward flow) along with the differences between the two periods are shown in figures 6a-f. Figures show an increase in temperature and salinity along the continental shelf under 100m, which is in agreement with the stronger transport/advection of warmer and saltier ESSW, poleward. The surprise is that negative "anomalies (differences between the two periods)" in temperature (Fig.6c) and salinity (Fig. 6f) are found in surface. I agree with the authors, this might result from the interannual/seasonal variations that have not been filtered from the signal analysed. However, this part of the discussion could be further elaborated (see general comment above), by (for example) looking at the SST from the satellites data from which intraseasonal variations can be estimated.

We agree with the reviewer that our discussion of temperature and salinity changes in the surface layer is very rudimentary; however, the main focus of our study is the circulation variability and its effect on hydrography and biogeochemistry. Given that section 4.3 and 4.4 are already quite long and detailed, as pointed out by the reviewer, we do not want to extend the analysis and interpretation of the surface signals.

**48) I411: Does N-loss means biogeochemical processes? Could the authors respecified which ones and clarify this sentence?**

Yes, the N-loss is a biogeochemical process that occurs within the ocean when there is no oxygen available (see e.g. Kalvelage et al., 2013). Bacteria use the oxygen atoms from nitrate ( $NO_3$ ) and nitrate ( $NO_2$ ) "to respire" and thus convert nitrogen nutrients to  $N_2$  gas while consuming organic matter (i.e. they reduce nitrate and nitrite). We have added a description of the individual anaerobic biogeochemical processes when discussing the nutrient distribution to make this more clear and state when first mentioning N-loss:

The increasing nitrogen deficit is caused by the microbially facilitated reduction of nitrate, nitrite and ammonium to  $N_2$  gas which occurs in anoxic waters during the consumption of organic matter (e.g. Kalvelage et al., 2013).

49) I412: Why would you expect this? Is the increase of Nitrate (and then, the reduce nitrogen deficit) not related to the stronger transport poleward in the PUS of high nutrient ESSW as shown by Echevin et al., 2014? You may want to re-specified (or show?) the mean Nitrate characteristics and provide the mean alongshore gradient of Nitrate to support your demonstration.

Indeed, the increase of nitrate is due to the advection of water with higher concentrations, but this spatial gradient is set by the biogeochemical N-loss. We state this more clearly now in the manuscript. In the T-S diagram we see an increase of nitrate for unchanged T-S conditions in the ESSW range, therefore the properties of the ESSW itself have changed.

Along the ESSW pathway within the PCUC from the equator to  $12^{\circ}$  S, the nitrogen deficit increases while nitrate concentrations decrease (Silva et al., 2009; Zamora et al., 2012; Kalvelage et al., 2013). The increasing nitrogen deficit is caused by the microbially facilitated reduction of nitrate, nitrite and ammonium to N2 gas which occurs in anoxic waters during the consumption of organic matter (e.g. Kalvelage et al., 2013). The resulting nitrogen deficit accumulates with time during the poleward advection.

**50) I423-425: the meaning is unclear**

We have changed the sentence to be clear.

However, the increase of nitrate (and the sum of inorganic nitrogen species) exceeds the phosphate decrease by a ratio higher than the nitrogen to phosphorus relation implied in equation 1, therefore the nitrate change is more important than the phosphate change for the nitrogen deficit reduction.

**51) I432: It won't be the case if equatorial waters were less rich in nutrients than the PUS. The sign of the anomaly depends on the sign of current anomaly and the sign of the gradient of the tracer (temperature or biogeochemical variables).**

Indeed this depends on the sign of tracer gradient. As seen in their supplementary material the study by Bachèlery et al. (2016) is done in a setting of poleward nitrate decrease as well, we have added a mention of this to the manuscript.

In a model study in the Atlantic Ocean, were nitrate decreases poleward as well, it was shown that the total effect of CTWs on nitrate concentrations varies regionally ...

**52) 1436: due to**

Changed.

53) I438: Do the authors see changes in the coastal ecosystem? I wonder if the nutrient input associated with the downwelling CTW and the change in the N-P ratio is associated with a phytoplankton bloom as describe in Echevin et al., 2014. Have the authors looked at satellites chlorophyll data?

Thank you for this comment. Satellite data suggest a continuous decrease of chlorophyll concentrations from April to June in agreement with the seasonal cycle. An increase, which may have been caused by the nutrient advection due to the CTW was not evident. However, the model results from Echevin et al. (2014) described the simultaneous propagations of several modes of which the fastest mode was associated with the SLA and the velocity signal while CTW modes responsible for productivity change were of higher order.

**54) I442- suggests**

Changed.

55) I447- Again, at least, the small differences observed in temperature, salinity and oxygen could simply be due to the fact that the seasonal and interannual variations cannot be removed from signal analysed. This statement is too strong for the conclusion.

Yes, thank you. We have weakened this statement to express that in principal, no change of properties due to advection is expected without the existence of a respective horizontal gradient of the property.

For parameters without strong horizontal gradients, an increase in PCUC flow does not cause pronounced changes in the advection. In this study this applies to the conservative properties temperature and salinity as well as for oxygen where alongshore gradients are weak (Zamora et al., 2012). For these parameters there are no large differences between both circulation phases that can be attributed clearly to the altered circulation. Yet concentrations of nutrients are influenced by shorter transit times, being less altered by biogeochemical cycling.

56) I449/451: I do not think that was shown here (or at least not pointed out effectively). This study does not show changes in the rate of N-loss but rather point out the stronger transport of nitrate as the mechanisms for the nitrate/nitrogen deficit anomaly (in line with the results of Echevin et al., 2014). To look at how the biogeochemical cycles are affected by the CTW propagation further analysis are required (for example the use of a model). I will rephrase this to make your point clearer.

We have changed the discussion to highlight our point that the amount of nitrate lost during the transport between the equator and 12 °S is lower for higher current velocities, because constant biogeochemical cycling rates cause less accumulated changes of concentrations during the shorter transit times.

**57) I452: "On intraseasonal timescales": From April to May 2017, our results suggest an increase in nitrate due to the passage of an intraseasonal downwelling CTW...**

We have changed the manuscript according to the suggestion.

For the period from April to May 2017, our study suggests an increase in nitrate levels due to the passage of an intraseasonal downwelling CTW. This contrasts with the decrease observed previously on interannual timescales caused by downwelling CTWs (Graco et al., 2017).

**58) I453: A downwelling CTW?**

We have added "downwelling" in the comparison to the study by Graco et al. (2017). See above in the answer to 57.

**59) I454 outcomes**

Changed.

**60) I458: different from**

Changed.

---

## Author Response (AR2)

**Reply to reviewer #2:**

**The authors conducted extensive field observations of flow, hydrographic conditions and the concentration of biogeochemical substances over the shelf and slope off Peru. They combined the satellite remote-sensed SLA and SST and the field observations to investigate the causes of the intra-seasonal variabilities of, for example, DO in the water column and qualitatively discussed the characteristics of the propagating CTW, and those of the variant types of nutrients and DO, as well as oscillation in OMZ. This manuscript is extensively condensed and improved from its previous OSD version with great improvement in the scientific focus. This manuscript is well-written, and the reflected sciences are well presented. I thereafter intend to suggest an "minor revision". However, since I still have serval suggestions for the authors to consider, I thereby suggest a major revision to let the authors think more in depth into the dynamics they proposed.**

We kindly thank the reviewer for the reevaluation of our manuscript.

In the manuscript we have improved the discussion of our results by focusing on our data and by adding a schematic of oxygen and nutrient changes. In the introduction we have reduced the number of cited studies and section 4.4 was reworked. We have improved the discussion as well by focusing on our own results. Additionally, we have checked the whole manuscript for typos and wrong numbering.

 In our detailed response below, comments by the reviewer are in bold letters and changes in the manuscript are expressed in italic letter.

**1) The authors revised the manuscript from their last submission to the OSD journal, and there are typos left. I suggest the authors to go through the entire manuscript very carefully.**

We thank you for pointing us at these issues, we made sure to check the manuscript for such typos.

**For example, L265, the last "maintaining" should be "maintain".**

Changed accordingly.

**L280, 6 month should be "months".**

Changed accordingly.

**L305, what does this "not inconsistent" mean? "consistent"? I didn't see any verb in this sentence.**

We agree that this sentence is not well written and reformulated it. Now we use "agrees" instead of the for statement "not inconsistent".

*Along the equator, the propagation speed of the SLA is about 2.7 m s$^{-1}$ (dashed line in the left panel of Fig. 7) which agrees with the phase speed of a first vertical mode equatorial Kelvin waves (e.g. Yu and McPhaden, 1999).*

**I stop here, but it is the duty of the authors to check the expressions sentence by sentence.**
**2) Too many citations in the introduction section. I suggest you to just preserve the most representative ones, and reduce the number of references substantially. Obviously, you didn't discuss all those studies.**

Thank you very much for this suggestion, accordingly we went through our introduction and restricted the citations to the most important ones for each statement.

**And I can not find Winkler 1888 (L130)?**

The publication by Winkler (1888) may be found using the doi-number included in the references (dx.doi.org/10.1002/cber.188802102122). Indeed, neither Web of Science nor Google Scholar find the article when searching for the title.

**3) L190, what does the exact meaning of the long sentences starting with "The data were …" I can understand that you did spatial smoothing but the description here is very confusing.**

We agree with you that this sentence is very long and therefore confusing, we have decided to shorten this sentence and restrict it to the most important information that smoothing was applied.

*Smoothed hydrographic and biogeochemical properties along the 12° S section were calculated by interpolating the data vertically onto common potential density surfaces. Averaging in density space removes the effect of signal smearing due to vertical displacement caused by internal waves. The profiles were then averaged in bins of 2 km according to distance from the coastline and then smoothed using 2-D Gaussian weights. In a final step, the averaged and smoothed sections were vertically transformed back into depth space using the smoothed depth – density relation.*

**4) You clarified the respective roles of wind stress curl and wind stress itself. In section 4.2.1. This is good, but could you please also show in the response (you may don't want to include them into the revised manuscript) about the role of tides? You don't need to discuss the flood-ebb tides, but spring-neap tides may play some roles?**

From mooring data of velocity collected in early 2013 at the same location we have calculated the alongshore amplitudes and phases of the O1, K1, N2, M2, S2 tidal constituents. From these we have reconstructed a 30 day timeseries of the alongshore velocities caused by the tides (Fig. 1).

[Figure]

Figure 1: Tidal alongshore velocities at three different water depths at the 12° S section off Peru.

The tides may introduce velocities of up to 5 cm s$^{-1}$ and the difference between neap and spring tide amplitude is about 2 cm s$^{-1}$. These velocities are small compared to the increase of poleward flow of more than 20 cm s$^{-1}$ over a large part of the PCUC depth range and therefore very unlikely to be important for the changes of PCUC strength.

**5) Still, section 4.4 is too descriptive. This is important, since that massive and systematic observations are new to us to understand the background and variabilities. However, it is still your obligation to help the readers to get insight into your science.**

According to your suggestion we have improved section 4.4 to explain better explain the concentration patterns of the nutrients and which biogeochemical processes are responsible for them. Overall, we made this section much less descriptive and connect changes of different parameters with each other more.

Please find the changed section 4.4 in the revised manuscript.

**I thereby strongly suggest the authors to include a schematic plot for different patterns of the CTWs propagation, changes in the shelf currents and the responses of the nutrients and DO you show in section 5.**

Following your suggestion, we have included a schematic of the oxygen and nutrient response to the circulation change.

[Figure]

*Figure 12: Schematic of the conditions off Peru during the initial phase of weak PCUC flow (a) and strong PCUC flow (b). Alongshore circulation is shown as colour shading, two isopycnals are sown in black and the top of the anoxic zone in magenta (represented by the 2 μmol kg⁻¹ isoline). The green shaded area in (b) shows the region of increased nitrate, decreased phosphate concentrations and decreased nitrogen deficit.*

**You cited too much of some others studies in the section 5, and some contents can be extracted from your own observation. I may suggest you to further condense and deepen the sciences in section 5 by your own data, although the current arrangement of logics looks sensible. This is the core of my major concern.**

In the discussion (section 5) we focus more on our results and include the aforementioned schematic for this as well. Additionally, we have removed parts of the discussion which were not directly focused on the results of our study but relying on a lot of cited literature.

Please find the changed section 5 in the revised manuscript.

**Reply to reviewer #3**

**I read the revised manuscript and the authors' responses to the comments. Overall, following both reviewers' comments, the authors have made adequate modifications to the deficiencies, especially the unclear explanation on physical processes, in the first draft. The revised paper have discussed the possible effects of various dynamic factors, including CTWs, local winds, topography, eddies etc., on the intraseasonal strengthening of PCUC. The comprehensive analyses of these potential factors indicate the essential role of CTWs in modulating the variability of sea level and PCUC. Although the authors has not carried out further numerical simulation, the existing data analysis results still provide a lot of valuable information on dynamic processes and physical-biogeochemical interactions. These interesting results are worthy of further quantification through the numerical experiments of coupling models in the future. At the same time, I also noticed that some new errors were introduced during the modification. Thus, I suggest that the authors read through the text and revise it carefully.**

We thank you very much for reviewing our manuscript.

In the manuscript we have improved the discussion of our results by focusing on our data and by adding a schematic of oxygen and nutrient changes. In the introduction we have reduced the number of cited studies and section 4.4 was reworked. We have improved the discussion as well by focusing on our own results. Additionally, we have checked the whole manuscript for typos and wrong numbering.

 In our detailed response below, comments by the reviewer are in bold letters and changes in the manuscript are expressed in italic letter.

**Detailed comments are the following:**

**1) The authors modified the order of some figures, but the text part was not completely revised accordingly. As a result, some figures do not match the corresponding discussions in the text. For example: l 284-288: Fig. 7 should be Fig. 6. The descriptions of Figures 8, 9 and 10 in Sections 4.3-4.4 actually correspond to figures 9, 10 and 11 in the list.**

We thank you for noting this and have corrected the figure referencing.

**2) Section 4.1.3 should be Section 4.2.3.**

Thank you very much for finding this mistake, we have changed the section numbering.

**3) P7, L272-274: "…… However, during the later period the strong acceleration of the poleward flow occurred, requiring an intensification of alongshore wind stress." It is difficult to imagine why the intensification of alongshore wind will accelerate the poleward undercurrent. To my understanding, given that the prevailing winds are southeasterly along the Peruvian coast, the weakening of local southeast trade winds will excite poleward-propagating downwelling CTWs, similar to those driven by the equatorial westerly winds. In fact, some studies on the coastal El Nino in spring 2017 have shown that the northerly coastal wind anomalies and the equatorial westerly winds anomalies are both important in maintaining the high sea level along the Peruvian coast (e.g. Peng et al., 2019).**
**Peng, Q., Xie, S., Wang, D. et al. Coupled ocean-atmosphere dynamics of the 2017 extreme coastal El Niño. Nat Commun 10, 298 (2019). https://doi.org/10.1038/s41467-018-08258-8.**

Indeed, this formulation is difficult to understand. An increase of alongshore wind stress will create upwelling waves reducing poleward transport, but this effect vanishes once the wave propagated poleward. Additionally, these waves set up an alongshore pressure gradient which is forcing the PCUC and persists until the wind forcing changes (Yoon and Philander, 1982).

We have changed the paragraph to be shorter and focus on the mismatch of wind stress and poleward flow which is therefore not the driven of PUC intensification. Therefore, we have removed the complicated and easily misunderstood explanation of the mechanisms of wind forcing from this section.

*On May 15 the wind stress increased and remained elevated for a period of about 5 days. This is not in agreement with the changes of poleward transport: The transport increased from late April to its maximum mid-May, whereas the alongshore wind stress in this period did first decrease and later increase.*

In the discussion we have included the study of Peng et al. when discussing the role of CTWs for the warming during the Coastal El Niño.

References:

[revised manuscript text omitted]